
# The Space CARBon Observatory (SCARBO) concept: Assessment of XCO₂ and XCH₄ retrieval performance

Matthieu Dogniaux[1], Cyril Crevoisier[1], Silvère Gousset[2], Étienne Le Coarer[2], Yann Ferrec[3], Laurence Croizé[3], Lianghai Wu[4], Otto Hasekamp[4], Bojan Sic[5], Laure Brooker[6]

[1]Laboratoire de Météorologie Dynamique/IPSL, CNRS, École polytechnique, Institut Polytechnique de Paris, Sorbonne Université, École Normale Supérieure, PSL Research University, 91120 Palaiseau, France
[2]Institut de Planétologie et d'Astrophysique de Grenoble, Université Grenoble-Alpes, 38058 Grenoble, France
[3]ONERA/DOTA, BP 80100, chemin de la Hunière, 91123 Palaiseau, France
[4]SRON Netherlands Institute for Space Research, Utrecht, the Netherlands
[5]NOVELTIS, 31670 Labège, France
[6]Airbus Defence and Space, 31 rue des Cosmonautes, 31402 Toulouse, France

*Correspondence to*: Matthieu Dogniaux (matthieu.dogniaux@lmd.ipsl.fr)

**Abstract.** Several single-platform satellite missions have been designed during the past decades in order to retrieve the atmospheric concentrations of anthropogenic greenhouse gases (GHG), initiating worldwide efforts towards better

monitoring of their sources and sinks. To set up a future operational system for anthropogenic GHG emission monitoring, both revisit frequency and spatial resolution need to be improved. The Space CARBon Observatory (SCARBO) project aims at significantly increasing the revisit frequency of spaceborne GHG measurements, while reaching state-of-the-art precision requirements, by implementing a concept of small satellite constellation. It would accommodate a miniaturized GHG sensor named NanoCarb coupled with an aerosol instrument, the multi-angle polarimeter SPEXone. More specifically, the

NanoCarb sensor is a static Fabry-Perot imaging interferometer with a 2.3 x 2.3 km² spatial resolution and 200 km swath. It samples a truncated interferogram at optical path differences (OPDs) optimally sensitive to all the geophysical parameters necessary to retrieve column-averaged dry-air mole fractions of CO₂ and CH₄ (hereafter XCO₂ and XCH₄). In this work, we present the Level 2 performance assessment of the concept proposed in the SCARBO project. We perform inverse radiative transfer to retrieve XCO₂ and XCH₄ directly from synthetic NanoCarb truncated interferograms, and provide their systematic

and random errors, column vertical sensitivities and degrees of freedom as a function of five scattering error-critical atmospheric and observational parameters. We show that NanoCarb XCO₂ and XCH₄ systematic retrieval errors can be greatly reduced with SPEXone posterior outputs used as improved prior aerosol constraints. For two thirds of the soundings, located at the centre of the 200 km NanoCarb swath, XCO₂ and XCH₄ random errors span 0.5 – 1 ppm and 4 – 6 ppb, respectively, compliant with their respective 1-ppm and 6-ppb precision objectives. Finally, these Level 2 performance

results are parameterized as a function of the explored scattering error-critical atmospheric and observational parameters in order to time-efficiently compute extensive L2 error maps for future CO₂ and CH₄ flux estimation performance studies.





## 1. Introduction

The monitoring of anthropogenic greenhouse gas (GHG) emissions is crucial to assess the progress made towards the 2015 Paris Agreement goals, and satellite remote estimations of GHG atmospheric concentration can help to better constrain anthropogenic and natural GHG emissions through top-down atmospheric inversion studies (Ciais et al., 2010). As urban areas account for about 70% of all fossil fuel related emissions on small areas (Duren and Miller, 2012), a frequent monitoring of local scale and point sources would enable to constrain a large fraction of anthropogenic carbon dioxide emissions. With near and shortwave (NIR and SWIR) infrared measurements, that are sensitive to atmospheric layers close to the surface where emissions take place, the spectro-imagery of $CO_2$ performed by large-swaths sensors with small ground-size adjacent pixels (e.g. 2x2 km$^2$) offers the adequate spatial resolution to detect point-source emission plumes. Anthropogenic emission rates can then be inferred by using Gaussian plume models (e.g. Bovensmann et al., 2010) or usual atmospheric flux inversion approaches (e.g. Pillai et al., 2016; Broquet et al., 2018). Coverage and revisit frequency are also critical for an operational emission monitoring system. For instance, considering satellites carrying sensors with a 250 km swath, the annual number of detected $CO_2$ plumes over Berlin ranges from 13 to 50 with a constellation that includes from 1 to 6 satellites, respectively (Kuhlmann et al., 2019). Five satellites are enough to ensure a daily global coverage at a fixed overpass time (Velazco et al., 2011).

Currently flying NIR and SWIR satellite missions include JAXA's Greenhouse gases Observing SATellites (GOSAT and GOSAT-2), NASA's Orbiting Carbon Observatory-2 and 3 (OCO-2 and OCO-3), the Chinese mission TanSat and ESA's Sentinel 5-Precursor/TROPOMI. $CO_2$ and/or $CH_4$ integrated columns are retrieved from their measurements thanks to inverse radiative transfer algorithms that determine the state of the atmosphere that best fit the infrared measurements provided by these missions. Imperfections in forward radiative transfer and inverse modelling result in systematic errors or increased variability of the retrieved GHG columns with regard to reference products, such as those produced by the ground-based Total Carbon Column Observing Network (TCCON) (Wunch et al., 2011). In NIR and SWIR spectral bands, taking into account scattering particles such as optically thin cirrus clouds or aerosols is particularly critical as they change the optical path of the measured radiation. Their imperfect modelling thus results in sizeable systematic errors of retrieved column-averaged dry-air mole fractions of $CO_2$ (denoted $X_{CO_2}$) (e.g. Houweling et al., 2005; Reuter et al., 2010). State-of-the-art $X_{CO_2}$ retrieval algorithms do account for the detrimental impact of scattering particles, however empirical corrections of their results that depend on aerosol parameters are still necessary (e.g. Guerlet et al., 2013; Reuter et al., 2017; O'Dell et al., 2018; Wu et al., 2018). The remaining (or not corrected) systematic errors can then perturb GHG atmospheric flux inversions, as shown in synthetic flux inversion studies (e.g. Chevallier et al., 2007; Pillai et al., 2016; Broquet et al., 2018). Thus, accounting for the scattering particle impact on satellite-based GHG column retrievals remains a key challenge for the performance of satellite missions.



In addition, none of these currently flying NIR and SWIR missions have the spatial coverage, spatial resolution nor revisit frequency that meet the requirements for operational top-down monitoring of anthropogenic GHG emissions. Compact GHG sensors concepts are well suited to address these previous limitations: their small sizes (and lower costs) allow to envision constellation concepts that could close the coverage and revisit gaps in the objectives of current or planned single-platform high-end reference instruments (e.g. Varon et al., 2018; Strandgren et al., 2020; Wilzewski et al., 2020). However, it requires

that their precisions reach an acceptable level of performance: better than 1 ppm and 10 ppb for $X_{CO_2}$ and $X_{CH_4}$ precisions, respectively, in the case of the upcoming high-end Copernicus $CO_2$ Monitoring (CO$_2$M) mission (Meijer and Team, 2019).

The Space CARBon Observatory (SCARBO, https://scarbo-h2020.eu/) project funded by the European Union Horizon 2020 research and innovation program investigates the feasibility of a low-cost GHG monitoring satellite constellation (Brooker,

2018). The proposed concept targets natural and anthropogenic GHG emissions, and aims to address the previously described limitations through various design features. First, SCARBO satellites would carry a miniaturized GHG sensor named NanoCarb (~9 kg), which is a static Fabry-Perot imaging spectrometer that samples truncated interferograms at optical path differences (OPDs) related to the GHG signature in NIR and SWIR spectral regions. These OPDs are selected to be optimally sensitive to geophysical parameters necessary to retrieve $X_{CO_2}$ and $X_{CH_4}$ (Ferrec et al., 2019; Gousset et al.,

2019). The currently considered imager would have a ~200 km swath with a 2.3 x 2.3 km$^2$ spatial resolution, enabling to detect emission plumes from megacities and hotspots (e.g. > 10 Mt CO$_2$ yr$^{-1}$ power plants). Secondly, the NanoCarb sensor would be coupled with an aerosol instrument, the multi-angle polarimeter SPEXone (van Amerongen et al., 2019; Hasekamp et al., 2019) whose measurements can help limiting the impact of scattering errors in GHG retrievals, and thus mitigating the systematic errors they can cause on $X_{CO_2}$ and $X_{CH_4}$ (Rusli et al., 2021). Both the NanoCarb and SPEXone instruments could

be carried on small satellite platforms (< 100 kg). With the objective of reaching sub-ppm and sub-6-ppb precisions for $X_{CO_2}$ and $X_{CH_4}$, respectively, a SCARBO constellation could thus be envisioned as a valuable companion to CO$_2$M. Finally, with about 20 satellites, it could provide daily revisits (and even intraday depending on the regions, with cloudy overpasses included) over megacities and emission hotspots and thus close the revisit gap in the current CO$_2$M plans. More specifically, the SCARBO project pursues two parallel objectives: (1) the development of an airborne prototype for the NanoCarb

concept that can be deployed in an airborne campaign together with the SPEX airborne instrument (Smit et al., 2019); (2) the performance assessment of the NanoCarb coupled to SPEXone concept for GHG column retrieval (Level 2, hereafter L2) and GHG flux estimation (Level 4, hereafter L4).

In this work, we present the Level 2 performance assessment of the concept proposed in the SCARBO project. For a set of

scattering error-critical atmospheric and observational parameters, we perform inverse radiative transfer to retrieve $X_{CO_2}$ and $X_{CH_4}$ directly from synthetic NanoCarb truncated interferograms. This differs from usual concepts that use infrared spectra as measurements: the discontinuous and sparse sampling of NanoCarb truncated interferograms do not allow to calculate the



spectra through Fourier Transform formalism. In this paper, we first seek to analyse the information content of such

measurements as well as their vertical sensitivities. Following the approach outlined in (Buchwitz et al., 2013) for the preparation of CarbonSAT, retrieved $X_{CO_2}$ and $X_{CH_4}$ systematic and random retrieval errors are then analysed and parameterized as functions of the explored atmospheric and observational parameters. We especially study the impact of improved prior knowledge of aerosol parameters brought by SPEXone measurements on the L2 performance of the concept. Finally, considering a synthetic constellation of SCARBO satellites, we exemplify how the derived L2 error parameterizations can be applied to time-efficiently compute large $X_{CO_2}$ and $X_{CH_4}$ error maps that can be used as inputs to L4

performance studies.

This paper is structured as follows: Section 2 describes the NanoCarb and SPEXone instruments. Section 3 presents the general approach and the inverse method used in this work. Section 4 details the synthetic atmospheric setup, as well as the two studied design scenarios: without and with SPEXone aerosol measurements that can be used as improved prior

constraint for GHG retrievals. Section 5 details NanoCarb measurement information content, the vertical sensitivity of the retrieved columns, and describes the two design scenario retrieval results. Section 6 presents the L2 error parameterization approach and illustrates how it can be applied to yield typical error maps. Finally, Section 7 highlights the conclusions of this work.

## 2. Description of the SCARBO concept

### 2.1 NanoCarb

NanoCarb is a static Fourier Transform imaging spectrometer that samples a truncated interferogram at optical path differences (hereafter OPDs) optimally sensitive to geophysical parameters necessary to retrieve $X_{CO_2}$ and $X_{CH_4}$. Its optical design, the optimized OPD selection, the measurement principles, the expected radiometric performance and the resulting statistical error on $X_{CO_2}$ are extensively described in (Gousset et al., 2019).


**Table 1. NanoCarb spectral band characteristics**

|  | Band 1: $O_2$ A-band | Band 2: $CO_2$-weak | Band 3: $CH_4$-band | Band 4: $CO_2$-strong |
|---|---|---|---|---|
| **Region** | 0.76 μm | 1.6 μm | 1.66 μm | 2.06 μm |
| **Measurement** | Surface pressure, aerosols | $CO_2$, $H_2O$ | $CH_4$ | $CO_2$, aerosols |
| **Narrow-band filter reference wavenumber** | 13,093 cm$^{-1}$ | 6,213 cm$^{-1}$ | 6,078 cm$^{-1}$ | 4,840 cm$^{-1}$ |

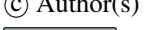



| | | | | |
|---|---|---|---|---|
| **Narrow-band filter FWHM** | 35 cm$^{-1}$ | 24 cm$^{-1}$ | 69 cm$^{-1}$ | 18 cm$^{-1}$ |
| **Radiative transfer simulation limits** | 12,940 – 13,175 cm$^{-1}$ | 6,180 – 6280 cm$^{-1}$ | 6,000 – 6200 cm$^{-1}$ | 4820 – 5010 cm$^{-1}$ |

To summarize, narrow-band filters, described by their central wavelength and full-width at half-maximum (FWHM), first select the light incoming from a given field of view (hereafter FOV) in the four spectral bands considered for the NanoCarb

instrument and detailed in Table 1. For each spectral band, the truncated interferogram is sampled thanks to an array of Fabry-Perot interferometers of fixed OPDs. They produce images of the whole FOV modulated with interference rings on the camera detector. Thus, an image of the FOV is recorded for each spectral band and for all of their respective selected OPDs, and, conversely, a truncated interferogram is measured at the selected OPDs for all the ground pixels within the FOV. Figure 1 shows how spectral bands, OPDs and FOV images are accommodated on the instrument detectors: the measured

intensity depends on the observed atmospheric scene, on the spectral band and OPD and also on the transversal $\theta_T$ (across-track) and longitudinal $\theta_L$ (along-track) angles characterizing a given ground pixel within the FOV. This spectral response at pixel-level arises from the angular dependence of the Fabry-Perot and narrow-band filter transmissions. Figure 2 shows a synthetic NanoCarb measurement corresponding to one of the central pixels of the NanoCarb FOV displayed in Fig. 1. Finally, the camera detector captures snapshots with a frequency set so that NanoCarb records a truncated interferogram, for

all the FOV ground pixels, every time the FOV moves forward by one ground pixel in the along-track direction.

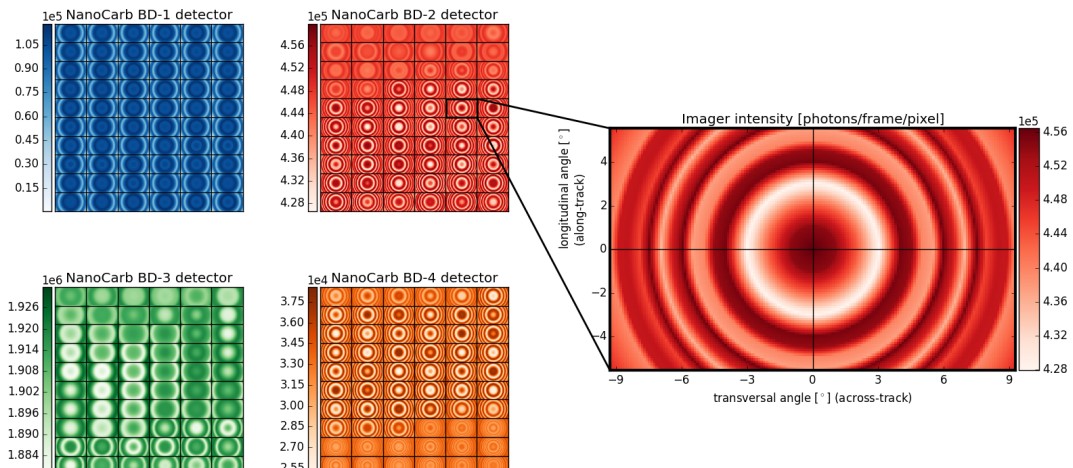





**Figure 1. Example of a complete NanoCarb measurement. Left panels show the measured intensities for the 4 spectral bands (denoted BD in the figure), for all their 60 OPDs and for all FOV pixels. The right panel illustrates the FOV intensity for one given OPD. This example has been computed for a vegetation surface type, with a solar zenith angle of 25°.**

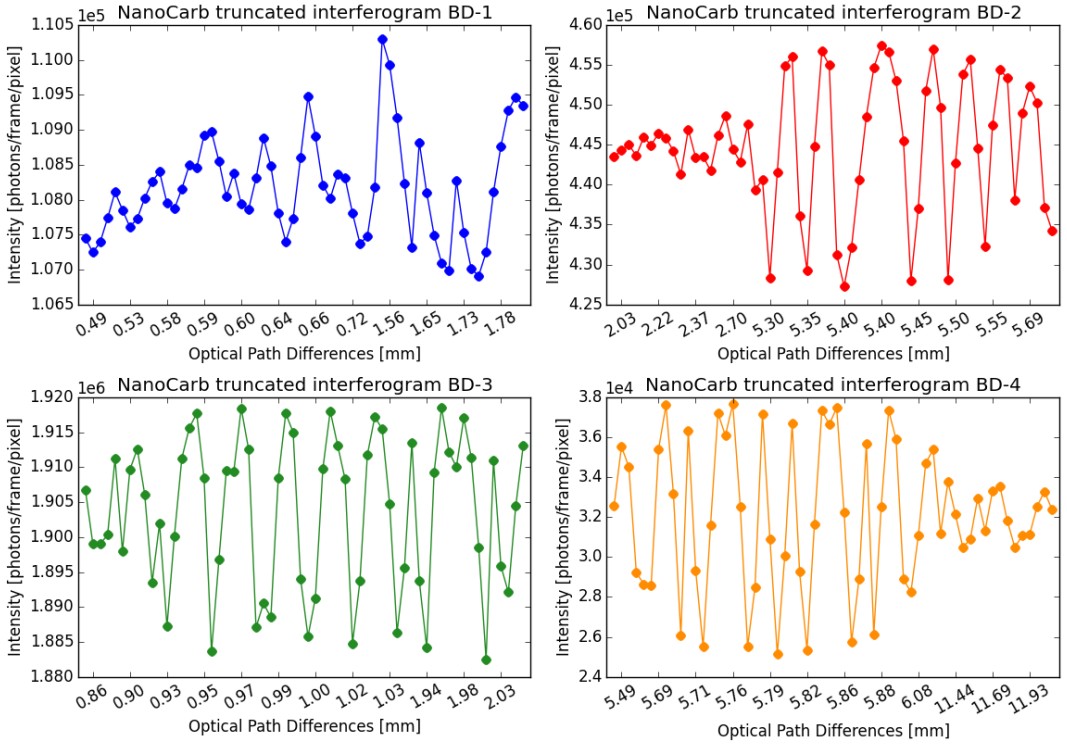


**Figure 2. Example of a NanoCarb truncated interferogram for the 4 spectral bands (denoted BD in the figure), for a central pixel of the field-of-view, simulated for the same situation as for Fig. 1. Intensities are plotted as a function of the OPD indices as the OPD sampling is very discontinuous.**

This work uses the latest design of the NanoCarb concept, based on an selection of 60 OPDs per spectral band, optimized for

the central part of a FOV that accommodates 170 (across-track, $\theta_T$ between -9.3° and 9.3°) x 102 (along-track, $\theta_L$ between -

5.5° and 5.5°) ground 1.15x1.15 km$^2$ pixels. Entanglements between $CO_2$, $CH_4$, $O_2$, $H_2O$ and aerosols signals have been

considered, with the assumption that albedo models are constant over all four spectral bands. Here, we use a NanoCarb

instrumental model that implements (1) a model of the spectral transmission for a 3-cavity narrow-band filter that simulates

the angular dependency within the FOV (2) an analytical approximation of the Fabry-Perot transmission (Gousset et al.,

2019). Given a synthetic radiance spectra, computed by a forward radiative transfer model at a pseudo-infinite spectral



resolution, and the transversal $\theta_T$ and longitudinal $\theta_L$ angles characterizing a given ground pixel within the FOV, it yields a NanoCarb truncated interferogram. This analytical model assumes perfect processing of the raw measurements and does not account for possible optical defects (such as straylight) or instrumental inaccuracies (such as faulty thermal regulation, which

impact is assessed separately), nor does it take into account the Point Spread Function (PSF) of the instrument.

### 2.2 SPEXone

SPEX (standing for Spectro-Polarimeter for Planetary Exploration) is a family of aerosol sensors that have been co-developed by the Netherlands Institute for Space Research (SRON) and its academic and industrial partners. SPEXone, the latest and most compact multi-angle polarimeter of this family (6 dm$^3$), is currently being developed by SRON, supported by

optical expertise from Airbus Defence and Space Netherlands and the Netherlands Organisation for Applied Optics (TNO) (van Amerongen et al., 2019). It measures visible light at 5 viewing angles ± 50°, ±20° and 0° along the satellite track and makes use of the spectral modulation technique (Snik et al., 2009) to encode the Degree of Linear Polarization (DoLP) in the measured spectrum. Radiance measurements will be provided at the spectral sampling (1 nm) and resolution (2 nm) of the spectrometer. The Degree of Linear Polarization (DoLP) will be provided at 50 spectral band with a spectral resolution

ranging from about 10-30 nm. A key feature of SPEXone is that it is designed to measure the DoLP at very high accuracy (0.003) allowing the retrieval of aerosol size, refractive index, and single-scattering albedo in addition to the Aerosol Optical Depth (AOD) (Hasekamp et al., 2019).

### 2.3 Sizing of the SCARBO constellation concept

The constellation sizing aims at ensuring intra-daily revisit of the largest possible amount of anthropogenic $CO_2$ emission hotspots which emission rate is compatible with the 1 ppm SCARBO $X_{CO_2}$ precision objective. For this purpose, we use the reprocessing of the Open Source Data Inventory of Anthropogenic $CO_2$ (ODIAC) database performed by Wang et al. (2019). They identified the emission clumps that are compatible with the detection of an $X_{CO_2}$ anthropogenic plume by a satellite flying around noon, for different $X_{CO_2}$ precision. Figure 3 shows the repartition of the emission hotspots compatible with the

1 ppm SCARBO precision objective, and gives the revisit statistics over these hotspots for a constellation of 22, 24 or 26 satellites flying at 600 km on Sun-Synchronous orbits, and equally distributed over two orbital planes, at 10:00 and 14:00. With 24 satellites, the SCARBO constellation provides a global coverage and guarantees a daily revisit for all hotspots, and intra-daily revisit for 73% of the hotspots (those beyond ±30° of latitude). This number of satellites compromise well between coverage, cost and available launch and deployment capabilities.



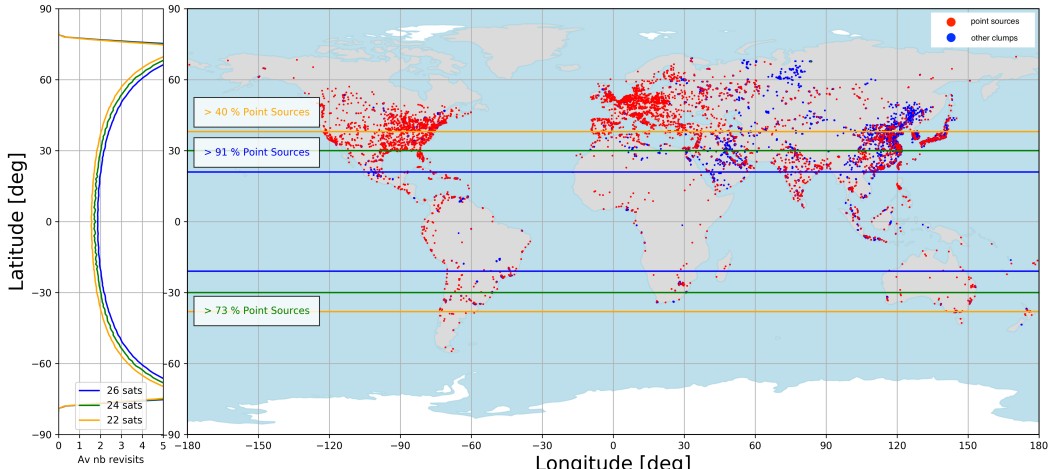


**Figure 3. Map of hotspots (point sources in red, and other type of emission clumps in blue) with an emission rate compatible with the detectability of an $X_{CO_2}$ plume around noon with a 1 ppm precision spaceborne instrument. Yellow, green and blue horizontal lines give the minimum latitude for better intra-day revisit for a constellation of 22, 24 and 26 satellites, respectively. The left panel gives the corresponding average number of revisits as a function of latitude for a constellation of 22, 24 and 26 satellites,**
**respectively.**

### 3. Methodology

#### 3.1 General approach

Level 4 atmospheric flux inversions targeting regional or global scale exploit extensive amounts of Level 2 products. The preparation of planned satellite missions usually includes Observing System Simulation Experiments (OSSEs) that build a
realistic numerical model of the atmosphere, simulate the instrument orbit and its measures, on which retrieval algorithms are then tested. This kind of computation-expensive approach is out of scope for the early-stage readiness of the NanoCarb truncated interferogram concept. This is why we propose here to follow the approach used for CarbonSat preparation (Buchwitz et al., 2013).

First, given synthetic atmospheric and aerosol models (Sect. 4.1), we introduce five scattering error-critical atmospheric and observational parameters (Sect. 4.2) for which we simulate synthetic NanoCarb truncated interferograms, without artificial noise, for parameter values that span realistic intervals. Then, for two different SCARBO design scenarios (without and with SPEXone, described in Sect. 4.3) we assess the L2 performance of the concept by performing inverse radiative transfer to

retrieve $X_{CO_2}$ and $X_{CH_4}$ directly from the previously simulated synthetic NanoCarb measurements. Key L2 performance

results presented in Sect. 5 comprise the systematic and random errors of the retrieved $X_{CO_2}$ and $X_{CH_4}$, as well as the vertical

sensitivities of these retrieved columns, which are, for instance, essential to yield pseudo-observed columns from simulated

GHG concentration profiles. Finally, those results are parameterized as functions of the selected scattering error-critical

atmospheric and observational parameters (Sect. 6). This yields fast and easily usable L2 performance models that enable to

produce large amounts of L2 data.

**3.2 Retrieval method: the 5AI inverse scheme**

In the context of this work, inverse radiative transfer aims to determine the geophysical parameters and their uncertainties

that best explain a given noised infrared measurement. For this purpose, we use the 5AI retrieval scheme (Dogniaux et al.,

2021) that implements the Optimal Estimation (hereafter denoted OE) inverse method (Rodgers, 2000).

In the framework of OE, we consider a state vector $\boldsymbol{x}$, containing various geophysical variables that adequately describe the

state of the atmosphere and of the surface, and a measurement vector $\boldsymbol{y}$ (here, the NanoCarb truncated interferogram). Both

are Gaussian random variables described by an average and an uncertainty given in a covariance matrix. Prior to the

measurement, climatologies or ad hoc choices describe the knowledge of the state $\boldsymbol{x}$: this is called the a priori state. Its

uncertainty has a strong impact on the retrieval result as it constrains how the measurement can be allowed to modify the

state. Given this a priori state with its uncertainty and the measurement $\boldsymbol{y}$, which uncertainty is known thanks to the noise

characteristics of the instrument, OE enables to find the most probable a posteriori state $\hat{\boldsymbol{x}}$ that best fit the measurement $\boldsymbol{y}$,

thus verifying the following equation:

$$\boldsymbol{y} = \boldsymbol{F}(\boldsymbol{x}) + \boldsymbol{\varepsilon} \tag{1}$$

with $\boldsymbol{F}$, the forward radiative transfer model that describes the physics linking the state to the measurement and $\boldsymbol{\varepsilon}$, the

measurement noise which statistics is known with the instrument and detector characteristics. Besides, OE relies on a

Bayesian formalism that translates the measurement uncertainty into state uncertainty, thus yielding an a posteriori

covariance matrix, for the retrieved a posteriori state $\hat{\boldsymbol{x}}$, that describe the estimation random error. Finally, OE also provides

the averaging kernel matrix, usually denoted $\boldsymbol{A}$, which describes how the retrieved state $\hat{\boldsymbol{x}}$ relates to the true (but unknown)

values of its chosen parameters.


The key L2 performance results that we seek to determine are computed from these outputs: (1) the systematic errors of the

retrieved $X_{CO_2}$ and $X_{CH_4}$ are defined as the differences between retrieved columns, computed from the a posteriori state $\hat{\boldsymbol{x}}$,

and the synthetic true columns (2) $X_{CO_2}$ and $X_{CH_4}$ random errors are computed from the a posteriori covariance matrix (3)

$X_{CO_2}$ and $X_{CH_4}$ vertical sensitivities are described by the column averaging kernels, computed from the averaging kernel

matrix $\boldsymbol{A}$.





Here, 5AI state vector includes all the main geophysical variables that impact shortwave infrared radiative transfer and may interfere with $X_{CO_2}$ and $X_{CH_4}$ retrieval. Table 2 describes the state vector, the a priori value of its elements as well as their prior uncertainties (no covariance is taken into account). The interfering impact of temperature has not been taken into

account for the latest optimized OPD selection used in this work, and is not considered in the state vector. In addition, except for the prior uncertainties of the aerosol optical depths, which depend on the design scenario we consider (see Sect. 4.3), all the prior uncertainties are purposefully large as we also aim to determine the information content of the NanoCarb truncated interferogram. Finally, the standard deviation of the instrument noise used for the 5AI retrievals can be calculated as:

$$\varepsilon_{i,j} = \sqrt{I_{i,j} + r_j^2} \tag{2}$$

with $\varepsilon_{i,j}$, the standard deviation of the a priori noise, for the $i$-th OPD of the $j$-th spectral band, $I_{i,j}$, the truncated interferogram intensity, for the $i$-th OPD of the $j$-th spectral band, and $r_j$, the readout noise of the spectral band camera detector.

**Table 2. 5AI retrieval state vector**

| Parameters | Size | A priori value | A priori uncertainty |
|---|---|---|---|
| H$_2$O profile scaling factor | 1 factor | 1.0 | 0.1 |
| CO$_2$ profile scaling factor | 1 factor | 1.0 | 0.1 |
| CH$_4$ profile scaling factor | 1 factor | 1.0 | 0.1 |
| Surface pressure | 1 | 1013.0 hPa | 4.0 hPa |
| Constant band-wise albedo | 4 spectral bands | Synthetic true value | 1.0 |
| Coarse mode aerosol Optical Depth (COD) | 1 layer | *Depends on design scenario* | *Depends on design scenario* |
| Fine mode aerosol Optical Depth (FOD) | 1 layer | *Depends on design scenario* | *Depends on design scenario* |


For its forward radiative transfer simulations, the 5AI scheme relies on the operational version of the Automatized Atmospheric Absorption Atlas (4A/OP) (Scott and Chédin, 1981) that is coupled with the LInearized Discrete Ordinate Radiative Transfer model (LIDORT: Spurr, 2002) in order to take into account multiple scattering caused by thin clouds and/or aerosols. Regarding spectroscopy, we use the 2015 version of the Gestion et Études des Informations

Spectroscopiques Atmosphériques: Management and Study of Atmospheric Spectroscopic Information (GEISA) spectroscopic database (Jacquinet-Husson et al., 2016) and we take into account line-mixing and collision-induced





absorption in the $O_2$ A-band (Tran and Hartmann, 2008), as well as line-mixing and $H_2O$-broadening of $CO_2$ lines (Lamouroux et al., 2010).

For this work, in order to take into account the use of truncated interferograms, 5AI is coupled to the NanoCarb instrumental model: for all the spectral bands defined in Table 1, a synthetic spectrum and its partial derivatives with regard to the state variables (also called Jacobians) are computed at pseudo-infinite spectral resolution by 4A/OP, and used as inputs to the NanoCarb instrumental model. It yields a NanoCarb truncated interferogram and its partial derivatives, which are the measure and its Jacobians used within the 5AI scheme in this work, respectively. As an example, Fig. 4 shows the partial

derivatives with regard to the 5AI state vector elements of the NanoCarb truncated interferogram shown in Fig. 2.

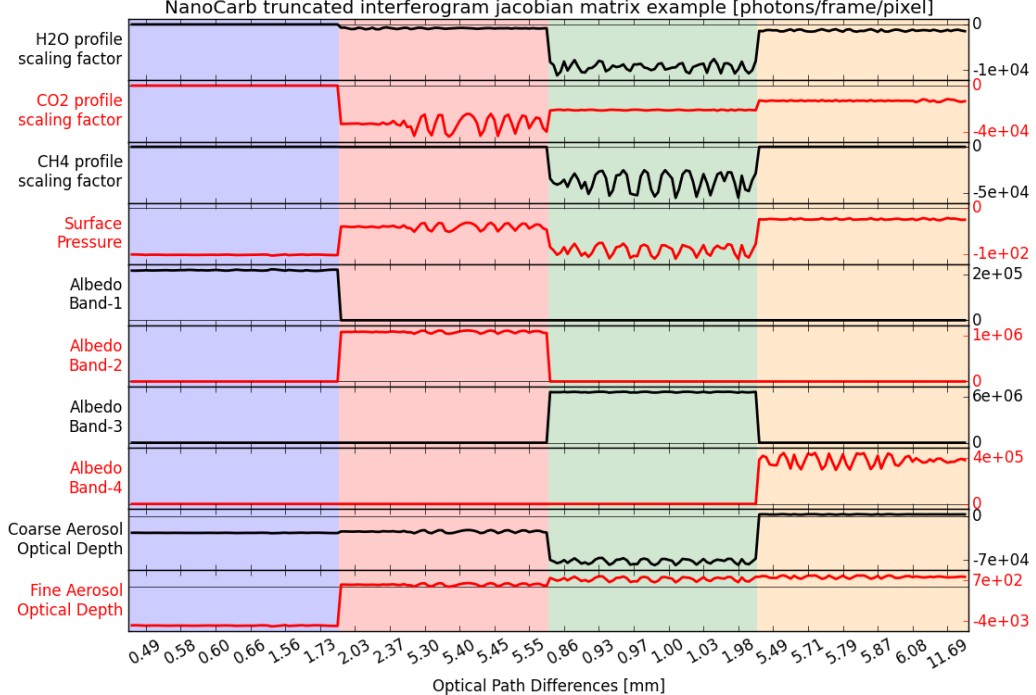

**Figure 4. Example of a NanoCarb truncated interferogram jacobian matrix, for the 5AI state vector geophysical variables used in this work. It corresponds to the NanoCarb truncated interferogram shown in Fig. 2. Again, partial derivatives are plotted as a function of the OPD indices as the OPD sampling is very discontinuous.**



### 3.3 Field of view

One NanoCarb measurement is not made of one truncated interferogram but of 170 (across-track) x 102 (along-track) truncated interferograms measured in snapshot mode, in the instrument field of view (FOV). As the foreseen time between two consecutive snapshots corresponds to the FOV moving by one ground pixel, up to 102 independent single-pixel NanoCarb truncated interferograms can be measured for a given fixed location on the ground, during the ~20-second overflight by the SCARBO satellite. The strategy to achieve the sub-ppm and sub-6-ppb precision objectives for $X_{CO_2}$ and $X_{CH_4}$, respectively, is then to combine, for every ground pixel associated with a transversal position $\theta_T$ within the swath, all their respective available along-track single-pixel measurements in order to retrieve one final unique state of the atmosphere per transversal position. Consequently, all the NanoCarb retrieval results presented in this work also depend on the transversal position $\theta_T$ of the situation within the swath.

Several hypotheses are made to fasten calculations within the FOV. Because we assume that it is uniform, it suffices to compute single-pixel L2 results for the whole FOV, and then to combine them in the along-track direction, in order to simulate the final L2 results (for details, see Appendix A). In addition, we assume that the along-track direction is aligned with the sun, single-pixel L2 results are thus perfectly symmetrical with respect to the longitudinal axis (results can be shown for positive values of $\theta_T$ only) and nearly symmetrical, because of the impact of the asymmetrical aerosol phase function, with respect to the transversal axis. Actually, due to the very nature of NanoCarb measurements (see interference rings in Fig. 1), single-pixel L2 results exhibit a near-central symmetry. Processing all 170 x 102 pixels within the FOV would lead to unmanageable computation times, this is why, here, we make use of the near-central symmetry in single-pixel L2 results to perform retrievals for a careful selection of 23 NanoCarb FOV pixels only (see supplements). Single-pixel L2 performance results are then interpolated from those 23 selected pixels to the whole FOV. Assuming that the CO₂ and CH₄ state vector parameters can be retrieved independently from the other geophysical variables, we then combine the single-pixel L2 performance results for $X_{CO_2}$ and $X_{CH_4}$ in the along-track direction following, for scalar quantities, the method described in Appendix A. This last step yields final L2 performance results that only depend on the transversal position $\theta_T$ within the swath, in addition to the five atmospheric and observational parameters considered here (see Sect. 4). Errors arising from the interpolation have been assessed and are negligible (not shown). These final L2 results are, like single-pixel L2 results, symmetrical along $\theta_T$, so only results for $\theta_T$ between 0° and 9.3° need to be shown.

## 4. Simulation setups

### 4.1 Synthetic atmospheric and aerosol models

The L2 performance assessment presented here is done for an atmospheric situation representative of the meteorological conditions that can be found over Europe. More precisely, for all our inverse radiative transfer simulations, we use the





average mid-latitude temperate atmospheric situation computed from the Thermodynamic Initial Guess Retrieval (TIGR) climatology library (Chedin et al., 1985) (available at https://ara.lmd.polytechnique.fr/index.php?page=tigr). The corresponding temperature, water vapour and ozone profiles have been discretized over 20 pressure levels bounding 19 layers, as for the ACOS algorithm (O'Dell et al., 2018). The surface pressure is set to 1013 hPa. For this synthetic

performance study, constant trace gas concentration profiles have been used: 394.85 ppm for $CO_2$ and 1855.3 ppb for $CH_4$.

We consider the presence of two aerosol modes in the atmosphere: a fine mode and a coarse mode. This assumption is in line with the ones made for the SPEXone retrieval capability study (Hasekamp et al., 2019) and, apart from the cirrus contribution, follows the assumptions made for the Full Physics retrieval algorithm developed at the University of Leicester

(Cogan et al., 2012). Here, the fine aerosol mode is treated under a log-normal size distribution with an effective size of 0.20 µm, an effective variance of 0.2 µm and a refractive index of $1.50+10^{-7}i$. This fine mode is representative of typical industrial non-organic aerosols and is located in a fixed atmospheric layer between 0 and 2 km. As for the coarse mode, it is treated under a log-normal size distribution with an effective size of 1.6 µm, an effective variance of 0.6 µm, a refractive index of 1.53+0.00254i and a spheroid fraction of 0.95. This coarse mode is representative of typical mineral dust and is

located at a varying altitude. The optical properties of these fine and coarse aerosol modes (extinction coefficient, single-scattering albedo and asymmetry parameter) are computed and used as inputs to the 5AI scheme.

### 4.2 Atmospheric and observational parameters for L2 performance assessment

We consider five parameters related to scattering error: (1) the albedo model (2) the solar zenith angle (3) the coarse layer height (4) the coarse mode aerosol optical depth (5) the fine mode aerosol optical depth. Those are usual parameters

considered for L2 performance assessments as they can strongly impact the photon optical path or the overall amount of signal measured by the satellite detector (Boesch et al., 2011; Buchwitz et al., 2013). Different values for these five parameters are explored, yielding a set of 324 atmospheric and observational situations for which the SCARBO L2 performance assessment is performed.

Regarding albedo (hereafter ALB), we consider three different ground albedo models representative of soil, vegetation and desert scenes, that are generated from the ASTER spectral library (Baldridge et al., 2009). As detailed in Sect. 2, the current optimization of the NanoCarb OPDs assumes constant band-wise albedos. Hence, in this work, the simulated NanoCarb truncated interferograms and the $X_{CO_2}$ and $X_{CH_4}$ retrievals use the same assumption: Fig. 5 shows the spectral dependence of the three albedo models we consider, as well as the constant band-wise fits used for all simulations.



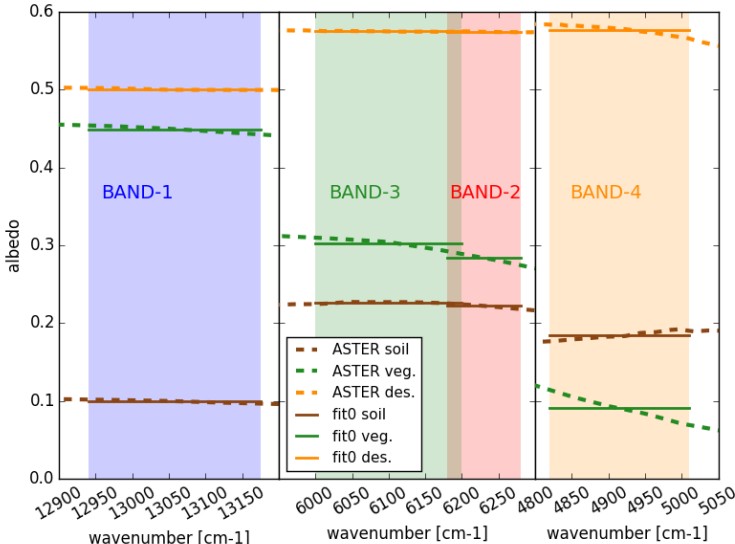


**Figure 5. True spectral dependence of the three albedo models considered in this work. Constant band-wise fits of these models are used here.**

For Solar Zenith Angle (hereafter SZA), we explore four different values: 0°, 25°, 50° and 70°. Though shortwave infrared

soundings can be made at higher SZAs, experience from OCO-2 post-filtering by the ACOS algorithm shows that soundings at high SZAs are more often removed (O'Dell et al., 2018), thus studying a maximum SZA of 70° is a reasonable compromise.

Concerning Coarse aerosol mode Layer Height (hereafter CLH), we assume possible altitudes of 2, 4 and 6 km and Coarse

aerosol mode Optical Depths (hereafter COD) explore the following values: 0.001, 0.05 and 0.15 at 550 nm reference wavelength. Fine aerosol mode Optical Depths (hereafter FOD) explore: 0.001, 0.12, 0.22 at 550 nm reference wavelength. The aerosol synthetic setup proposed here aims to represent: (1) background aerosol optical depth, arbitrarily attributed to industrial non-organic aerosols (as those are expected around and downwind of strong emission hotspots) with optical depth values consistent for instance with MODIS observed averages over Europe for 2010 (Palacios-Peña et al., 2019); (2)

transient coarse mineral desert dust layers that can be observed over Europe in late-spring, summer and early-autumn with a varying altitude (Papayannis et al., 2008).



### 4.3 Two design scenarios: without and with SPEXone

Two SCARBO satellite design scenarios are studied in this work. Table 3 summarizes the assumptions made for both scenario: they are only related to the a priori setups of 5AI NanoCarb retrievals.


**Table 3. Summary of no-SPEX and with-SPEX design scenario assumptions**

| Parameter | Prior | no-SPEX scenario | with-SPEX scenario |
|---|---|---|---|
| **Coarse aerosol mode Optical Depth (COD)** | A priori value | 0.05 | Synthetic truth |
| | A priori uncertainty | 0.5 | SPEXone linear error analysis output (see Fig. 6) |
| **Fine aerosol mode Optical Depth (FOD)** | A priori value | 0.12 | Synthetic truth |
| | A priori uncertainty | 0.5 | SPEXone linear error analysis output (see Fig. 6) |
| **Coarse Layer Height** *(not retrieved)* | A priori value | 2 km | Synthetic truth |

The first one, hereafter referred as 'no-SPEX', simulates a SCARBO satellite only carrying the NanoCarb instrument. This scenario is simulated with fixed a priori values for COD and FOD in the state vector, and with a fixed CLH of 2 km,
whatever the atmospheric and observational situation considered. The random prior uncertainties for COD and FOD are set to 0.5, a large value also reflecting the limited knowledge of aerosol parameters in this design scenario.

The second design scenario, hereafter referred as 'with-SPEX', simulates a SCARBO platform carrying both SPEXone and NanoCarb instruments at the same time, thus yielding collocated SPEXone and NanoCarb measurements. For this scenario,
we consider a two-step L2 retrieval approach in which SPEXone measurements are analysed first. These results are then used to improve the a priori constraints on aerosol parameters in second-step GHG column retrievals from NanoCarb measurements. The first step is fulfilled by a linear error analysis that yields SPEXone posterior uncertainties for COD and FOD, following the method described in (Hasekamp et al., 2019). Figure 6 shows these SPEXone random errors for the coarse mode (COD) and fine mode (FOD) aerosol optical depths, for all the 324 atmospheric and observational situations
considered in this work. Its first five top panels have a descriptive purpose: they remind the values of ALB, SZA, CLH, COD and FOD parameters for all the 324 situations. Thus, the first third of the x-axis is dedicated to soil albedo situations, the second to vegetation albedo situations, and the last to desert albedo situations. For all these ALB cases, all SZA values are explored, as are all scattering particle cases for all ALB cases and SZA values, thus sorting all the 324 considered





situations along one dimension (an identical sorting is used in Fig. 7, 9, 10 and 11). Regarding SPEXone performance, the
optical depths posterior uncertainties are correlated to the optical depths values and are lower for the fine mode compared to
the coarse mode. Uncertainties are higher for desert albedo situations as the ratio between scattered photons and surface-
reflected photons is more disadvantageous over desert compared to soil or vegetation situations. For both modes they
improve with increasing SZA values because the light path through the aerosol layer increases but also because more
favourable scattering angles are typically encountered at high SZA (Fougnie et al., 2020). Posterior uncertainties of coarse
mode optical depths are also decreasing with CLH values as more photons are scattered when the coarse layer height
increases. For the with-SPEX design scenario considered here, these COD and FOD posterior uncertainties are used as a
priori uncertainties within the second-step GHG column retrievals from NanoCarb measurements. In addition, in the absence
of full SPEXone retrieval results, we also assume that the first-step SPEXone measurement analyses yield perfectly accurate
COD and FOD values, as well as the true synthetic CLH values (aerosol layer heights are not retrieved from NanoCarb
measurements, as per Table 2, but can be obtained from SPEXone retrievals).

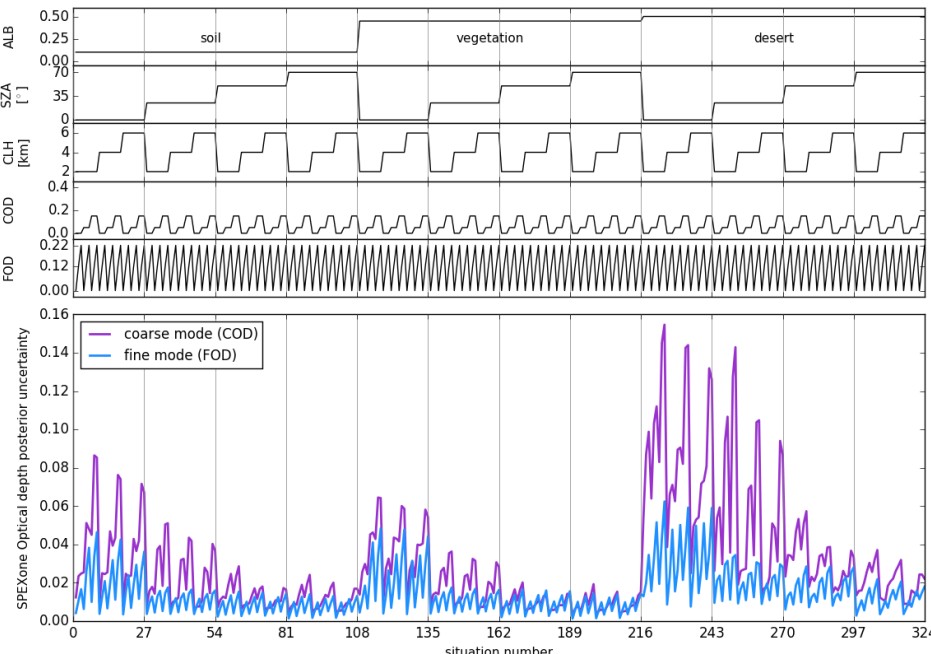

**Figure 6. SPEXone linear error analysis results for all the 324 atmospheric and observational situations used in this work.
SPEXone COD and FOD posterior uncertainties are plotted against the situation number: the five top panels detail the ALB (0.7
µm), SZA, CLH, COD and FOD values defining all these 324 situations.**



**5. Results and discussion**

**5.1 Geophysical information content and variable entanglements in NanoCarb truncated interferograms**

In this work, $X_{CO_2}$ and $X_{CH_4}$ are directly retrieved from truncated interferograms sampled at OPDs optimally sensitive to $CO_2$, $CH_4$ and possibly interfering geophysical variables. This peculiar nature of NanoCarb measurements strongly differs from usual infrared spectra (measured for example by GOSAT or OCO instruments). A way to evaluate the geophysical

information content is to examine the Optimal Estimation Degrees Of Freedom (hereafter denoted 'DOFs') that provide, for all state vector variables, the amount of useful independent quantities provided by the measurement, whatever its nature (Rodgers, 2000).

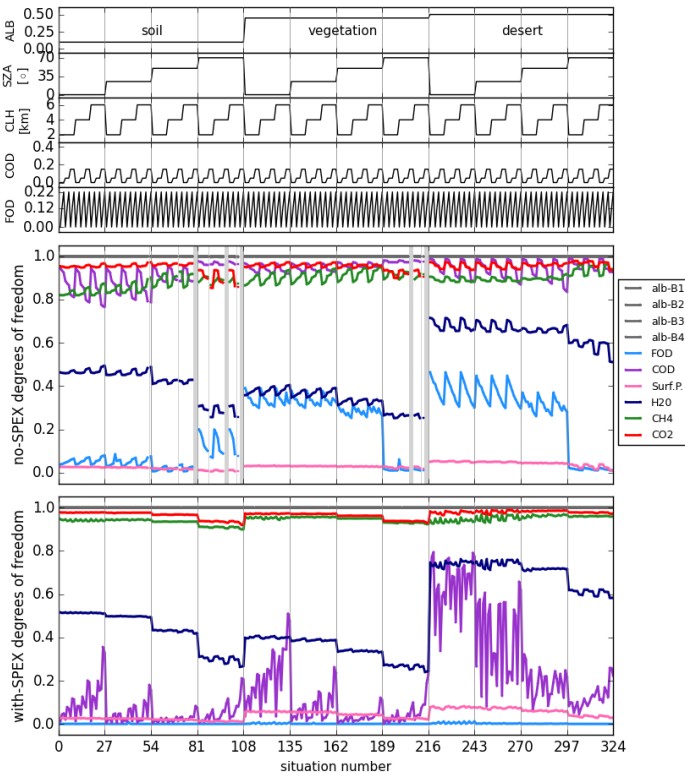

**Figure 7.** NanoCarb state vector variable degrees of freedom averaged over the 23 FOV pixels used to interpolate L2 results to the
whole FOV. Results are plotted as a function of the situation number: the five top panels detail the ALB (0.7 µm), SZA, CLH, COD and FOD values defining all the 324 situations. Grey-shaded areas denote situations for which retrievals did not satisfactorily converge.



Figure 7 shows NanoCarb state vector DOFs, averaged over the 23 selected FOV pixels, for all 324 atmospheric and observational situations considered in this work (described in the five top panels, as in Fig. 6), as well as for both no-SPEX and with-SPEX design scenarios. Grey-shaded areas in the no-SPEX design scenario case remove situations for which the retrievals did not satisfactorily converge. Overall, we can notice that $CO_2$ and $CH_4$ DOFs are close to 1.0, confirming the sensitivity of the retrievals and of NanoCarb measurements to these two target greenhouse gases. In the no-SPEX design

scenario, COD DOFs have similar values, thus underlying a significant sensitivity of NanoCarb measurement to the coarse mode aerosol layer. All albedo bands have 1.0 DOFs and, for other variables comprising water vapour profile scaling factor, surface pressure and FOD, retrievals do not get much information from NanoCarb measurements. This means that the retrievals rely on their a priori information for these variables, which can result in systematic $X_{CO_2}$ and $X_{CH_4}$ biases if the a priori is biased compared to the true state of the atmosphere. For the no-SPEX design scenario case, the DOFs evolution is

mainly explained by the strong sensitivity of NanoCarb measurements to COD. This COD sensitivity increases for situations with SZA=70° because spaceborne measurements are more sensitive to scattering for highly slanted optical paths. Conversely, this explains the drop of the other variable DOFs for which less measurement information is available in situations with SZA=70°. For all geophysical variables but albedo and surface pressure, the variations of DOFs are correlated with COD: the large 0.5 a priori uncertainty for COD in no-SPEX retrievals brings only a mild constraint that

results in the COD parameter driving the information content for all the other variables.

The with-SPEX design scenario exhibits much reduced aerosol parameter DOFs arising from NanoCarb interferograms: this scenario is designed so that SPEXone, with improved a priori constraints in GHG retrievals, brings much of the information regarding aerosols parameters. Consistently with SPEXone performance shown in Fig. 6, FOD DOFs are nearly equal to 0

thanks to SPEXone performance and the NanoCarb measurement mild sensitivity to fine mode aerosols. As for coarse mode aerosols, the remaining COD DOFs are the result of the strong sensitivity of NanoCarb measurements to this mode, and of SPEXone lower performance for coarse mode: with-SPEX COD DOFs are well correlated to Fig. 6 SPEX posterior uncertainties for COD. One can also note that, for similar SPEXone performance between fine and coarse at high SZAs in soil and vegetation case, COD DOFs are much larger than FOD DOFs, thus once again underlying the sensitivity of

NanoCarb measurement to the coarse mode. For low SZAs desert albedo situations, where SPEXone performance for coarse mode is at its lowest with large remaining uncertainties, COD DOFs are high, meaning that NanoCarb measurements can contribute to constraining coarse mode aerosols on these situations. Symmetrically to reducing the amount of NanoCarb measurement information used to constraint aerosol parameters, the use of SPEXone posterior results in NanoCarb GHG retrievals helps to use more of that information to constrain other variables. Consequently GHG, surface pressure and albedo

DOFs increase in the with-SPEX (with regard to no-SPEX) scenario, as shown in Fig. 7. This underlines the geophysical information entanglement of the latter variables with aerosol parameters in NanoCarb measurements.





Retrieving a profile scaling factor for $CO_2$ or $CH_4$ instead of a layered profile has the advantage of setting a 1.0 limit to the DOFs these gases can have. Given the state vector used in this work, reaching this 1.0 DOF limit value for all geophysical

variables would mean that all of them could be retrieved independently from each other. Failing to do so as shown in Fig. 7 for the with-SPEX scenario means that the geophysical information is entangled in NanoCarb measurements: variables cannot be retrieved independently from each other, correlations exists. A way to identify main variable-to-variable entanglements is to examine similarities (correlation or anticorrelations) between the partial derivatives of state vector elements. For example, Fig. 4 displays a correlation between albedo and $CO_2$ jacobians in NanoCarb band 2: both evolve

similarly around different continuous components. Though less or not visible due to scale, similar similarities exist between surface pressure and albedo Jacobians in band 1 (anticorrelation), $CH_4$ profile scaling factor and $H_2O$ or albedo Jacobians in band 3 (correlations) and $CO_2$ profile scaling factor and albedo Jacobians in band 4 (anticorrelation). Thus $CO_2$, albedo and aerosol variables are entangled in the current OPD optimization of NanoCarb measurement, and the same is true for $CH_4$ information, which is also entangled with $H_2O$.

**5.2 Vertical sensitivities: column averaging kernels**

Column averaging kernels (hereafter referred as 'AKs') describe the vertical sensitivity of retrieved $X_{CO_2}$ and $X_{CH_4}$. In other words, they show which atmospheric layers contribute the most to the GHG information contained in the measurement. NIR and SWIR spectrum measurements are typically sensitive to the whole atmospheric column, with AKs that reach their maximum in atmospheric layers close to the surface and then decrease with altitude above the mid-troposphere (e.g. for

OCO-2 Boesch et al., 2011). Figure 8 presents the NanoCarb $X_{CO_2}$ and $X_{CH_4}$ AKs for all albedo models and SZAs, and for the minimum and maximum total aerosol optical depth (AOD) situations. As for usual NIR and SWIR concepts such as OCO-2, or S5-P/TROPOMI, NanoCarb truncated interferograms are sensitive to $CO_2$ and $CH_4$ of all atmospheric layers. In addition, it can be noticed that NanoCarb AKs with low total AOD satisfactorily compare with those obtained for trace gas profile scaling factors retrieved from SCIAMACHY low-resolution measurements by the WFM-DOAS algorithm

(Bovensmann et al., 1999; Buchwitz et al., 2005). Indeed, like WFM-DOAS $X_{CO_2}$ AKs, NanoCarb $X_{CO_2}$ AKs grossly evolve from $1.2 - 1.5$ in the boundary layer to only $0.1 - 0.2$ at the top of the atmosphere (TOA), and the same comparison stands for $X_{CH_4}$ AKs: both evolve from approximately 1.2 in the boundary layer to about 0.5 at TOA. SZA dependence of AKs appears to be quite similar between NanoCarb and SCIAMACHY/WFM-DOAS for $X_{CH_4}$, but is different for $X_{CO_2}$ AKs: sensitivity in the boundary layer and lower troposphere decreases with SZA in NanoCarb case whereas it increases for

WFM-DOAS. Regarding NanoCarb AKs for atmospheric situations with the maximum aerosol optical depth, we can notice a sensitivity drop in the atmospheric layers containing aerosols for $X_{CO_2}$ AKs, especially at high SZAs. Even if comparing AKs shapes remain difficult because a $CO_2$ profile is retrieved (and not a scaling factor) and many other factors differ, a similar behaviour was noticed during the ACOS algorithm characterization (Boesch et al., 2011). As for NanoCarb $X_{CH_4}$


AKs, they exhibit a slight increase of sensitivity in the atmospheric layers containing aerosols, and a sensitivity drop

comparable to NanoCarb $X_{CO_2}$ AKs for SZA=70°. No similar $X_{CH_4}$ AKs sensitivity study to the presence of aerosol has been

found when writing this article.

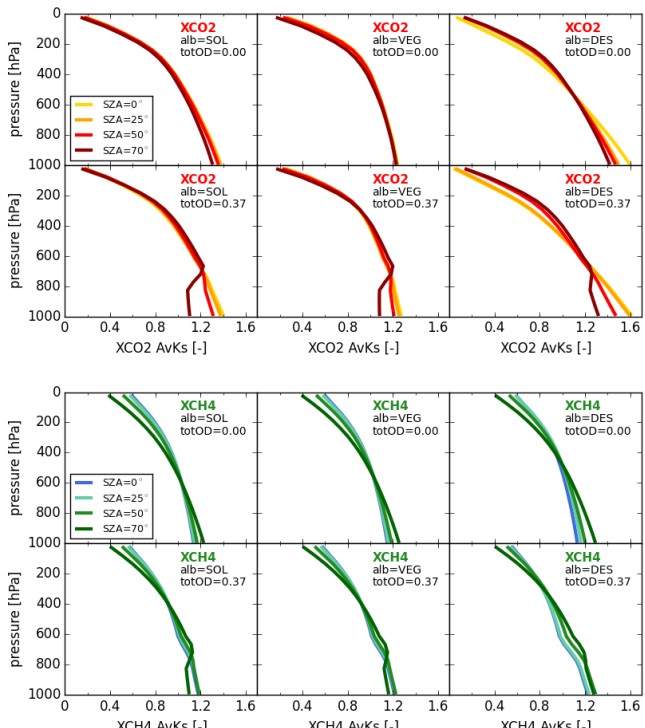

**Figure 8.** NanoCarb $X_{CO_2}$ (top panels) and $X_{CH_4}$ (bottom panels) column averaging kernels averaged over the 23 FOV pixels used
to interpolate L2 results to the whole FOV. We show all three albedo models: soil (left), vegetation (middle) and desert (right), and
four different SZA values (colour scales). Averaging kernels are shown for a low total aerosol optical depth (top rows) and a high
total aerosol optical depth (bottom rows).

### 5.3 Systematic and random errors

The NanoCarb spectral band narrow-band filters exhibit FOV-dependent effects that impact the L2 performance: their

reference wavelengths shift towards slightly shorter wavelengths with the angle of incident light, thus with the distance of

pixels to the centre of the FOV (Smith, 2008). As the OPD selection was optimized for the center of the FOV, this results in

an increased GHG and albedo information entanglement close to the swath border (not shown). This leads to an increase of

$X_{CO_2}$ and $X_{CH_4}$ random errors, which can become slightly larger than the SCARBO sub-ppm and sub-6-ppb precision

objectives for few situations and FOV pixels. In addition, it also challenges the hypothesis of independent scalar columns

that is used to combine the FOV single-pixel $X_{CO_2}$ and $X_{CH_4}$ results in the along-track direction. As a consequence, we

choose here to only consider pixels with $\theta_T$ comprised between -6° and 6° (whereas the full swath spans ±9.3° in the currently considered design used for constellation sizing).

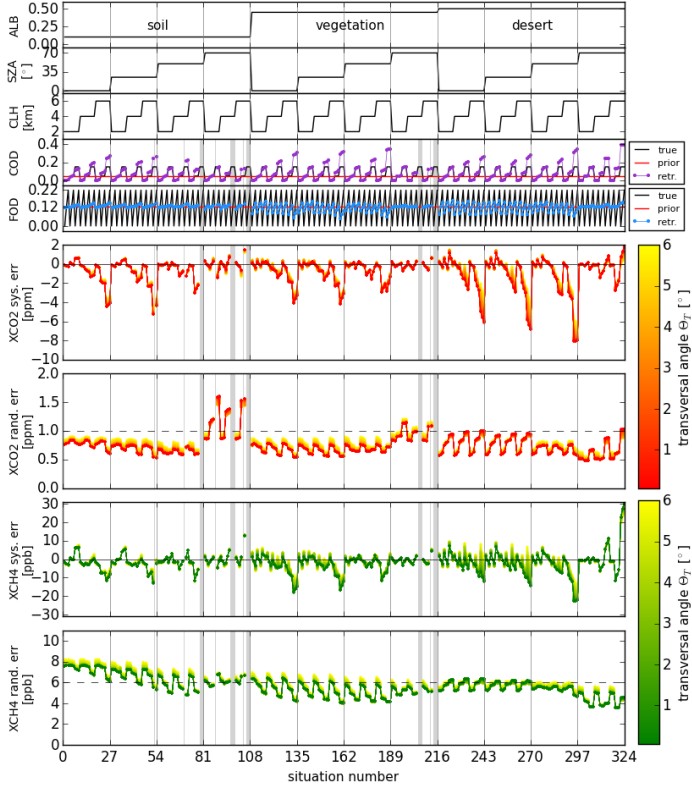

**Figure 9.** NanoCarb systematic and random errors for $X_{CO_2}$ and $X_{CH_4}$ in the no-SPEX design scenario case. Results are plotted as a function of the situation number: the five top panels detail the ALB (0.7 µm), SZA, CLH, COD and FOD values defining all the 480 324 situations. In addition, the a priori and averaged retrieved COD and FOD values are also shown in the 4th and 5th top panels. 6th and 7th panels show $X_{CO_2}$ systematic and random errors, respectively, and 8th and 9th panels show $X_{CH_4}$ systematic and random errors, respectively. The dependence on the transversal angle $\theta_T$ of L2 results is shown with the colour-scales: darker colours correspond to lower $\theta_T$ absolute values. Grey-shaded areas denote situations for which retrievals did not satisfactorily converge.


Figure 9 shows, for all 324 atmospheric and observational situations, as well as for all transversal angle positions between 0° and 6°, NanoCarb systematic and random errors (which definitions are given in Sect. 3.2) for $X_{CO_2}$ and $X_{CH_4}$ in the no-SPEX design scenario case. As for Fig. 6, the five top panels describe ALB, SZA, CLH, COD and FOD values for all situations. A





priori and retrieved values are also shown for COD and FOD, in order to explain where $X_{CO_2}$ and $X_{CH_4}$ systematic errors

come from. The four bottom panels display $X_{CO_2}$ and $X_{CH_4}$ systematic and random errors. Retrievals converge and satisfactorily reduce the cost function for most of the situations. Still, some of them remain challenging depending on the albedo model and SZA, when COD or CLH are far from the a priori value. Their results are excluded as shown with the grey-shaded areas.

In this no-SPEX case, systematic errors come from the erroneous prior knowledge of scattering parameters in the state vector (Fig. 9). Regarding scattering particles, NanoCarb measurements are mostly sensitive to the presence of coarse mode aerosols on the optical path (as explained in Sect. 5.1) and the COD can be retrieved to some extent when the synthetic truth is not too far from the a priori state. Retrieved FOD seldom differ from the a priori value, showing again that no-SPEX retrievals are very little sensitive to fine mode aerosols. Here, $X_{CO_2}$ and $X_{CH_4}$ systematic errors can reach up to 8 ppm and 30

ppb in absolute value for $X_{CO_2}$ and $X_{CH_4}$, respectively. This corresponds to about 10 times and 5 times their average random error, respectively. Thus, no-SPEX NanoCarb $X_{CO_2}$ retrievals are more sensitive to scattering error than no-SPEX NanoCarb $X_{CH_4}$ retrievals. $X_{CO_2}$ systematic errors are mostly driven by COD retrieval errors that correlate with CLH a priori misknowledge (CLH a priori value is here fixed at 2 km, see Table 3) and SZA. This SZA dependence of systematic errors is particularly important for $X_{CO_2}$: that may be explained by the use of the 2.05 µm $CO_2$ strong band that includes saturated

$CO_2$ lines and is quite sensitive to aerosols. A similar COD retrieval error dependence is found for $X_{CH_4}$ systematic errors, which also interestingly exhibit a stronger correlation to FOD retrieval error, particularly visible in vegetation and desert albedo situations. $X_{CO_2}$ and $X_{CH_4}$ systematic error swath dependence is shown by the colour scales. It is most visible when systematic $X_{CO_2}$ errors are high at the swath centre, and in situations with COD values far from the a priori for $X_{CH_4}$.

Random errors in the no-SPEX design scenario (Fig. 9), for transversal positions below 6° in absolute value, outperform the 1 ppm SCARBO $X_{CO_2}$ precision objective for SZA below 50° in soil and vegetation situations, and for all SZA values in desert albedo situations. Regarding $X_{CH_4}$ random errors, they overall meet the 6 ppb precision objective, but for soil albedo situations with SZA values of 25° or lower. Within the OE formalism random error variations are by definition completely correlated with DOFs variations (see Fig. 6): when more information is available for a given variable, its random error

diminishes. Thus, as for DOFs, random error variations are mostly driven by COD and ALB values in the no-SPEX design scenario. It can finally be noted that most of the transversal positions within the swath exhibit similar random errors, except for those close to 6°.

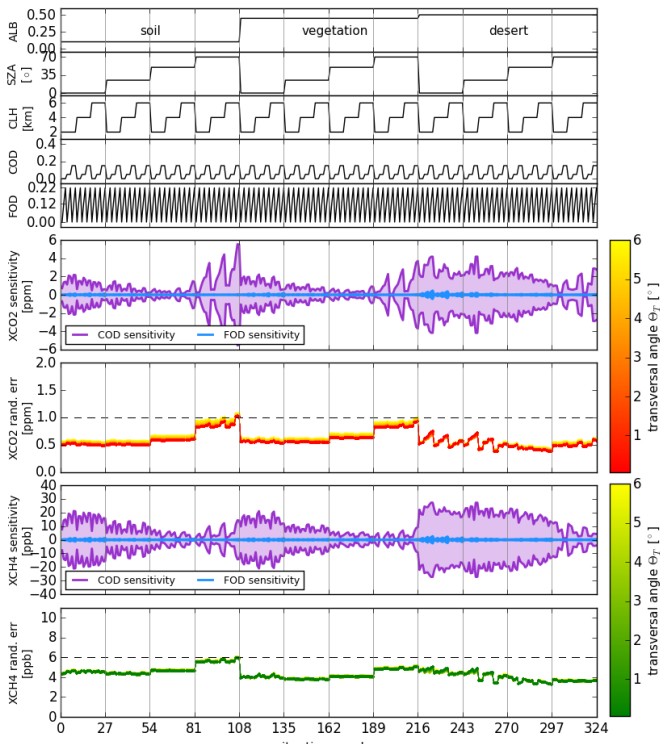

**Figure 10.** NanoCarb sensitivities to a priori misknowledge of COD and FOD and random errors for $X_{CO_2}$ and $X_{CH_4}$ in the with-SPEX design scenario case. Results are plotted as a function of the situation number: the five top panels detail the ALB (0.7 µm), SZA, CLH, COD and FOD values defining all the 324 situations. 6th and 7th panels show $X_{CO_2}$ sensitivities to a priori misknowledge of COD and FOD and random errors, respectively, and 8th and 9th panels show $X_{CH_4}$ sensitivities to a priori misknowledge of COD and FOD and random errors, respectively. The dependence on the transversal angle $\theta_T$ of random errors results is shown with the colour-scales: darker colours correspond to lower $\theta_T$ absolute values. As for sensitivities, only the minimum and maximum values for all transversal angles are shown.

As detailed in Table 3, the a priori aerosol profile and a priori state vector are identical to the true state of the synthetic atmosphere in the with-SPEX design scenario case. This is done to simulate a more precise and accurate knowledge of aerosol parameters that can be brought by the SPEXone instrument. Given this strong hypothesis for SPEXone retrieval accuracy, L2 retrieval results do not exhibit systematic errors. Consequently, in order to study the sensitivity of $X_{CO_2}$ and $X_{CH_4}$ systematic errors to this hypothesis, we use the averaging kernel matrix $\boldsymbol{A}$ to propagate a priori misknowledge of aerosol parameters. Following Rodgers' (2000) notations, we have:

$$d\hat{x} = \frac{\partial \hat{x}}{\partial x_{true}} dx_{true} = \boldsymbol{A} \, [0,0,0,0,0,0,0,0, \delta_{COD}, \delta_{FOD}]^T \qquad (3)$$





with $\hat{x}$, the retrieved state vector, $x_{true}$, the synthetic true state of the atmosphere, $\delta_{COD}$ and $\delta_{FOD}$ differential perturbation of
COD and FOD parameters, respectively.

Figure 10 is similar to Fig. 9, but shows results for the with-SPEX design scenario (a version that combines Fig. 9 and Fig.
10 is included in the supplements for a better comparison of systematic errors, but less readability of no-SPEX results).
Instead of systematic errors, it presents $X_{CO_2}$ and $X_{CH_4}$ systematic error sensitivities to synthetic truth perturbations of COD
and FOD corresponding to their respective prior uncertainties $\sigma_{COD}$ and $\sigma_{FOD}$: $\delta_{COD} = \pm\sigma_{COD}$ and $\delta_{FOD} = \pm\sigma_{FOD}$ (provided
by SPEXone linear error analysis). This systematic error sensitivity test is conservative in different ways. First, the
perturbation by SPEXone random error is at least a factor two greater than the systematic optical depth errors found in
(Hasekamp et al., 2019). In addition, the separation between COD and FOD perturbations does not allow for these errors to
compensate themselves and possibly partially cancel out a fraction of $X_{CO_2}$ and $X_{CH_4}$ systematic errors. $X_{CO_2}$ and $X_{CH_4}$
systematic error sensitivities to synthetic truth perturbations of COD and FOD are shown as the maximum and minimum
sensitivities among all transversal positions with $|\theta_T| < 6°$. Uncertainties in COD retrieved by SPEXone can result in up to
$\pm5.5$ ppm impact on $X_{CO_2}$ and $\pm28$ ppb impact on $X_{CH_4}$. It is interesting to note that despite similar SPEXone precisions for
COD and FOD between SZA=50° and SZA=70° over all albedo models, COD perturbations have a much more important
impact on $X_{CO_2}$ at SZA=70°. This highlights the particular sensitivity of the NanoCarb measurements to coarse mode
aerosols at high SZAs. This remains valid for $X_{CH_4}$ to a lesser extent. Sensitivities to COD imprecisions also impact $X_{CO_2}$
and $X_{CH_4}$ at low SZA over desert-albedo situations, where SPEXone uncertainties are the highest. Regarding fine mode,
uncertainties in FOD retrieved by SPEXone can result in up to $\pm0.4$ ppm impact on $X_{CO_2}$ and $\pm2.5$ ppb impact on $X_{CH_4}$.
Those sensitivities to FOD perturbations are greatly smaller than those to COD, due to the better SPEXone performance for
fine mode aerosols and the lower impact this mode has on NanoCarb measurements. Compared to the no-SPEX systematic
errors presented in Fig. 9, we can conclude here that SPEXone has the potential to significantly reduce systematic errors
originating from fine mode aerosols in both $X_{CO_2}$ and $X_{CH_4}$ retrievals from NanoCarb truncated interferograms. Regarding
coarse mode aerosols, the potential of SPEXone is more nuanced. SPEXone has COD posterior uncertainties at their best for
situations in which no-SPEX $X_{CO_2}$ retrievals exhibit the largest systematic errors due to COD, namely for high SZA values, in
situations with large COD. Conversely, no-SPEX $X_{CO_2}$ systematic errors are lower than the $X_{CO_2}$ impact of SPEXone COD
uncertainty in situations with low SZA. Thus, SPEXone performance for COD appears to be complementary to the $X_{CO_2}$
COD sensitivity of NanoCarb measurements. Considering a typical European situation with a vegetation albedo and
SZA=50°, the aerosol information brought by SPEXone is thus critical to reduce systematic errors due to coarse mode
aerosols. However, some situations where SPEXone is less precise for COD can remain a challenge: in case of transient
coarse aerosol contamination over desert albedo situations and low SZAs for instance. The sensitivity of $X_{CH_4}$ systematic
errors to SPEXone COD uncertainty is mostly larger than the no-SPEX $X_{CH_4}$ systematic errors, except for high SZA and





COD values in vegetation and desert albedo situations. This shows the limitations of SPEXone ability to help reduce the systematic errors originating from coarse mode aerosols in $X_{CH_4}$ NanoCarb retrievals.

Figure 10 also shows $X_{CO_2}$ and $X_{CH_4}$ random errors for the with-SPEX design scenario, those are lower than in the no-SPEX
design scenario. Indeed, due to the GHG and aerosol information entanglement shown in Sect. 5.1, the better a priori constraint of aerosol parameters brought by SPEXone enables to dedicate more of NanoCarb measurement information to estimate GHG parameters in the with-SPEX scenario. For nearly all the atmospheric and observational situations considered in this work, $X_{CO_2}$ and $X_{CH_4}$ satisfactorily reach the SCARBO sub-ppm and sub-6-ppb precision objectives, respectively.

## 6. Level 2 performance parameterization

**6.1 Linear regressions**

In order to yield generalized SCARBO L2 performance models from the 324 situations considered in this work, we adopt the approach used in (Buchwitz et al., 2013) and perform linear regressions to parameterize L2 performance results. In other words, we determine the $c$ and $a_i$ coefficients so that:

$$Y = c + \sum_{i=1}^{n} a_i X_i \tag{3}$$

with $Y$, an L2 performance result to parameterize as a function of $n$ heuristically determined (linear and non-linear) parameters $X_i$, expressed as combinations of the selected ALB, SZA, CLH, COD, FOD parameters along with $\theta_T$, the transversal angle position (absolute value) within the swath. Considering ALB_NIR, ALB_SWIR-1 and ALB_SWIR-2 that describe albedo model values at 0.7 μm, 1.6 μm and 2.0 μm, respectively, Table 4 lists the $X_i$ parameters used for $X_{CO_2}$ and $X_{CH_4}$ systematic errors, random errors and AK level values parameterizations.


**Table 4. Heuristically determined parameters to use for L2 performance parameterizations**

| Parameter name | Parameter definition for systematic errors ($n = 9$) | Unit to use |
|---|---|---|
| $X_1$ | $1/\cos{(SZA \times \pi/180)}$ | ° |
| $X_2$ | ALB_SWIR-2 | - |
| $X_3$ | $\log{(FOD)}$ | - |
| $X_4$ | $\log{(COD)}$ | - |
| $X_5$ | $\max(CLH, 2)$ | km |
| $X_6$ | $1/\cos{(\theta_T \times \pi/180)}$ | ° |
| $X_7$ | $(X_5 - 2) \times X_4$ | - |
| $X_8$ | ALB_NIR | - |




| Parameter name | Parameter definition for random errors ($n = 9$) | Unit to use |
|---|---|---|
| $X_9$ | $X_6 \times X_4$ | - |
| $X_1$ | $1/\cos(\text{SZA} \times \pi/180)$ | ° |
| $X_2$ | ALB_NIR | - |
| $X_3$ | $(-\text{ALB\_SWIR-2} + 0.2)$ | - |
| $X_4$ | $\log(\text{FOD})$ | - |
| $X_5$ | $\log(\text{COD})$ | - |
| $X_6$ | $1/\cos(\theta_T \times \pi/180)$ | ° |
| $X_7$ | $X_6 \times X_6$ | - |
| $X_8$ | $X_3/X_1$ | - |
| $X_9$ | $X_1 \times X_6$ | - |

| Parameter name | Parameter definition for column averaging kernel layer values ($n = 8$) | Unit to use |
|---|---|---|
| $X_1$ | $1/\cos(\text{SZA} \times \pi/180)$ | ° |
| $X_2$ | ALB_NIR | - |
| $X_3$ | ALB_SWIR-1 | - |
| $X_4$ | FOD | - |
| $X_5$ | COD | - |
| $X_6$ | $\max(\text{CLH}, 2)$ | km |
| $X_7$ | ALB_SWIR-2 | - |
| $X_8$ | $X_1 \times X_5$ | - |

Figure 11 shows parameterization results for $X_{CO_2}$ systematic and random errors, and for no-SPEX and with-SPEX design scenarios. For no-SPEX situations, results are parameterized for retrieved COD + FOD < 0.25, in order to emulate some sort

of sensible filtering that could be performed in operational processing. For systematic errors, the parameterization captures the combined COD, SZA and CLH trends of L2 results, as well as some of the transversal angle position dependence. Regarding no-SPEX random errors, the parameterization captures the combined albedo and SZA trends (except for soil-albedo at SZA=70°), but fails to reproduce the COD trend over vegetation and soil-albedo situations, due to the strong influence of this trend in desert situations. The transversal angle position dependence is well captured. The parameterization

for the with-SPEX scenario random errors is quite accurate and satisfactorily reproduces most of the L2 performance trends (for this scenario, we only filter with COD<0.6 and FOD<0.6). In addition, it can be noted that vegetation-albedo situations, the most representative of European surface, are those for which parameterizations best reproduce the computed exact L2 performance. Similar results are obtained for $X_{CH_4}$ (see supplements, where AKs parameterization results are also shown).





Overall, the means and standard deviations of parameterization approximation errors evaluated on all 324 situations and

transversal angle positions that passed filters are given in Table 5.

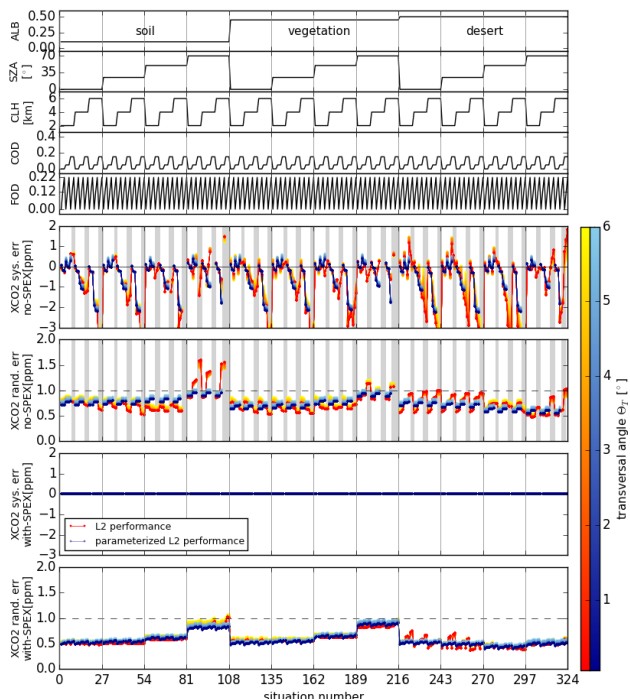

**Figure 11.** Parameterized (blue colour-scale) NanoCarb $X_{CO_2}$ systematic (6[th] and 8[th] panels) and random errors (7[th] and 9[th] panels) compared to exact L2 error retrieval results (red colour-scale). Results are plotted as a function of the situation number: the five top panels detail the ALB (0.7 μm), SZA, CLH, COD and FOD values defining all the 324 situations. Grey-shaded areas denote
situations for which retrievals did not satisfactorily converge or situations filtered out according to the retrieved COD and FOD values.

**Table 5. Parameterization approximation errors**

| Variable | Design scenario | Systematic error | Random error |
|----------|----------------|------------------|--------------|
| $X_{CO_2}$ | no-SPEX | $0.00 \pm 0.36$ ppm | $0.00 \pm 0.11$ ppm |
|  | with-SPEX | - | $0.00 \pm 0.05$ ppm |
| $X_{CH_4}$ | no-SPEX | $0.00 \pm 2.04$ ppb | $0.00 \pm 0.46$ ppb |
|  | with-SPEX | - | $0.00 \pm 0.24$ ppb |



### 6.2 Application of L2 performance parameterizations: 1st of July, 2015 example

SCARBO ground tracks are calculated and auxiliary datasets are gathered to provide large spatial and temporal scale maps
of the five selected error-critical parameters: ALB (at 0.7 μm, 1.6 μm and 2.0 μm), SZA, CLH, COD and FOD. We then use those maps to apply the previously obtained L2 performance parameterizations and yield systematic and random $X_{CO_2}$ and $X_{CH_4}$ errors, as well as $X_{CO_2}$ and $X_{CH_4}$ column averaging kernels, that can them be used for L4 flux inversion studies.

The SCARBO constellation considered in this study for the ground track computation is composed of 28 satellites on sun-
synchronous orbits of 605.498 km height, and separated on two orbital planes: one observing at 10 am (local time) and the second at 2 pm (local time). Orbital parameters are adjusted to have a repeating cycle of 7 days, and so that the second plane repeats the ground traces of the first one. As the provided L2 performance results already include the contribution of all along-track NanoCarb measurements, observations are sampled at the resolving spatial resolution of ~2.3 km in the across-track direction, producing 85 soundings in a 200 km swath corresponding to transversal angle positions $\theta_T$ between 0° in 9°
in absolute value.

Only clear sky land observations are kept: cloud flagging is performed with the MODIS Atmosphere L2 Cloud Mask Product (Ackerman, S. A., Frey, 2015) and land/sea flagging with the Global Multi-resolution Terrain Elevation Data (GMTED2010) 30'' product (Danielson, J.J., and Gesch, 2011). Given the date and time, the derived observation
geolocations enable to yield the SZA dataset.

Aerosol parameters COD, FOD and CLH are generated using the T255 Copernicus Atmospheric Monitoring Service (CAMS) reanalyses for aerosols (Flemming et al., 2017) interpolated at 15'' resolution. The different aerosol types proposed in this product are separated in two classes, a fine mode and a coarse mode, according to their overall size. Coarse mode and
fine mode optical depths (COD and FOD) datasets are then generated by summing the optical depths of the individual types belonging to each class. Finally, CAMS vertical mixing-ratios of aerosol types classified as coarse are processed to yield the average mass altitude that is used for the CLH dataset.

In order to create a ground albedo dataset at our three reference wavelengths, we employ the ESA ADAM (A surface
reflectance Database for ESA's earth observation Missions) climatology (Bacour et al., 2020) that relies on MODIS surface reflectance data. In order to extrapolate reflectance values at our three reference wavelengths, the Étude CLImatologique des Propriétés optiques de fonds de Sol (ECLIPS) French ANR project data is used (mentionned in Bacour, 2019), finally yielding ALB_NIR, ALB_SWIR-1 and ALB_SWIR-2 parameter datasets.





In order to illustrate the application of the L2 performance parameterizations, we use the parameter datasets of the 1st of July 2015 to compute parameterized $X_{CO_2}$ systematic and random errors for the 10 am (local time) orbital plane satellites, for both no-SPEX and with-SPEX design scenarios. Figure 12 shows 0.2°x0.2° averaged ALB_NIR, ALB_SWIR-2, SZA, CLH, COD and FOD cloud-free parameter maps. Unsurprisingly, albedo values are mostly representative of vegetation models (see Fig. 5), with rather high reflectance near 0.7 µm and low reflectance near 2.0 µm. Southern Spain and Italy, as well as

Maghreb have more desert-like surface albedos. For the 1st of July, 2015, CAMS simulated aerosols are mostly present over Maghreb, Eastern Spain, France, United-Kingdom and Eastern Europe, with rather high fine mode optical depth and low coarse mode optical depths. These coarse mode aerosols have a rather low layer altitude, except over Germany, where a low-COD layer reaches nearly 5 km. Figure 13 shows the corresponding parameterized $X_{CO_2}$ systematic and random errors for both design scenarios (see supplements for $X_{CH_4}$). Those are computed for transversal angle positions lower than 6° in

absolute values, as we do not consider the full NanoCarb swath in this work. Soundings are filtered with COD + FOD < 0.25 and COD < 0.6 and FOD < 0.6 in no-SPEX and with-SPEX scenario cases, respectively, thus explaining the different number of soundings between these two scenarios. No-SPEX systematic errors mostly correlate with COD and CLH as already noted in Fig. 7. whereas no systematic error is given for with-SPEX scenario as per its hypotheses. Regarding random errors, they are lower in the with-SPEX scenario and decrease over more desert-like albedo situations as can be seen

in Southern Spain and Italy as well as Maghreb.

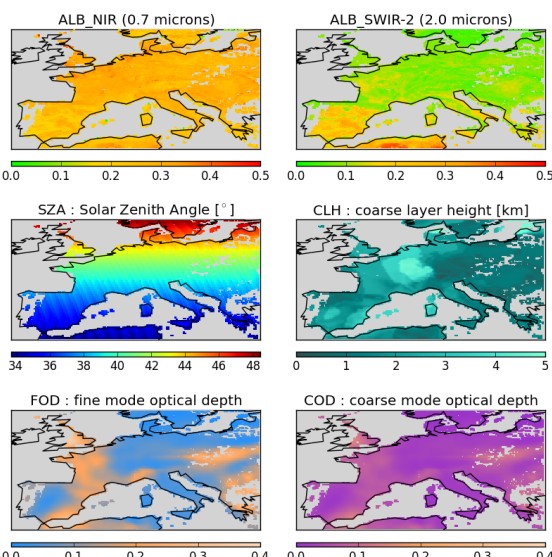

**Figure 12. ALB_NIR, ALB_SWIR-2, SZA, CLH, FOD and COD cloud-free parameter maps of the 1st of July 2015, averaged on a 0.2°x0.2° grid.**



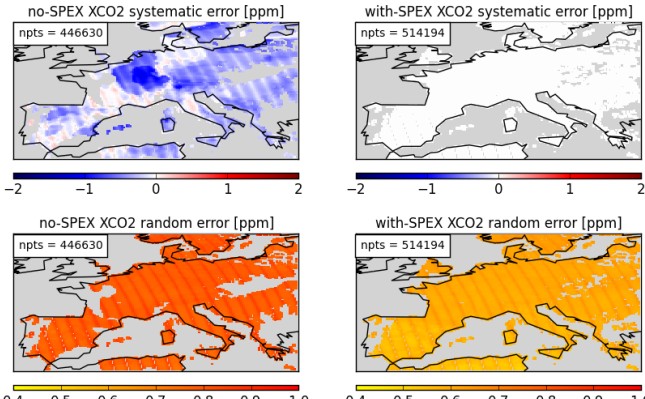

**Figure 13. Parameterized $X_{CO_2}$ systematic (top) and random (bottom) errors for the 1st of July 2015, for the no-SPEX (left) and with-SPEX (right) design scenario cases and averaged on a 0.2°x0.2° grid.**

## 7. Conclusions

In this work, we have carried out the Level 2 performance assessment of the NanoCarb concept developed in the SCARBO project. For a set of 324 scattering error-critical atmospheric and observational situations, we retrieved $X_{CO_2}$ and $X_{CH_4}$

directly from NanoCarb truncated interferograms by using the 5AI inverse scheme.

First, as this concept constitutes an original approach to NIR and SWIR infrared measurements compared to state-of-the-art GHG satellite missions, we have analysed the vertical sensitivities and information content of the truncated interferograms. Retrievals are clearly sensitive to $CO_2$ and $CH_4$ with degrees of freedom close to 1.0, and the retrieved $X_{CO_2}$ and $X_{CH_4}$ are

representative of all atmospheric layers as usual NIR and SWIR concepts.

In order to establish the merits of coupling NanoCarb with the SPEXone instrument dedicated to aerosols, we have compared the results for two SCARBO satellite design scenarios: no-SPEX and with-SPEX. Systematic $X_{CO_2}$ and $X_{CH_4}$ retrieval errors originating from the presence of fine mode aerosols on the optical path can be significantly reduced by taking

advantage of NanoCarb coupling with SPEXone. In addition, the performance of SPEXone for coarse mode aerosols also enables to reduce systematic $X_{CO_2}$ errors where they are the largest, for typical European vegetation albedo situation with SZA=50° and high COD for instance. Desert situations with low SZAs may still remain a challenge in case of transient desert dust contaminations for instance.



Regarding precision, $X_{CO_2}$ and $X_{CH_4}$ random errors span $0.5 - 1$ ppm and $4 - 6$ ppb, respectively. Thus, for transversal angle positions lower than 6°, NanoCarb are compliant with the 1-ppm and 6-ppb precision objectives for $X_{CO_2}$ and $X_{CH_4}$, respectively, for situations with SZA ≤ 50°.

These systematic and random retrieval column errors, as well as their vertical sensitivities, have been successfully
parameterized as functions of the five selected scattering error-critical parameters. Consequently, large L2 maps can be produced and distributed for L4 atmospheric GHG flux inversion performance assessments.

This simulation study sheds light on the Level 2 performance of the peculiar NanoCarb truncated interferogram concept: it exhibits an interesting potential for providing meaningful information about greenhouse gas atmospheric concentrations,
with a very compact imaging spectrometer. As for all simulation studies, there are implicit hypotheses that need to be considered: only scattering error-critical situations have been considered and prior knowledge of the true synthetic state of the atmosphere and of the surface is assumed to be perfect, but for aerosol parameters in the no-SPEX scenario. In particular, the number of aerosol types, their optical properties and number of layers are considered to be exactly known. In addition, the instrumental model is also ideal: it implements the theoretical Fabry-Perot interferometer equations without considering
any miscalibrated optical defect. However, as the SCARBO concept (NanoCarb coupled with SPEXone) reaches and even out-performs its precision objectives in this work with ideal hypotheses, we can expect some margins to cover for possible instrumental parameter imprecisions. This would be a next step towards an ultimately complete error budget that takes into account the critical instrumental (L1) and retrieval setup (L2) parameters that impact the overall performance of the SCARBO concept.


This first step in assessing NanoCarb L2 performance has also enabled to point out geophysical variable information entanglements in NanoCarb truncated interferograms, when examining the retrieval degrees of freedom. These include entanglements between albedo and surface pressure, albedo and $CO_2$, albedo and $CH_4$ and finally $CH_4$ and $H_2O$ that had not been taken into account for the NanoCarb optimized OPD selection and model used in this work. Because of the very nature
of NanoCarb measurements, these entanglements also evolve within the FOV, leading to an increase of $X_{CO_2}$ and $X_{CH_4}$ random errors on the swath edges. This specificity impacts the achievable swath for a given precision objective. It is consequently critical for the design of the SCARBO constellation, which results from a compromise between the number of satellites, the coverage and revisit possibilities. Thus, by identifying the limitations to disentangled GHG sensitivity within NanoCarb truncated interferograms, this work has also paved the way for future improvements of the whole concept design.





**Appendix A**

**Combining NanoCarb measurements in the along-track dimension**

With its current design, the NanoCarb instrument can gather up to $n = 102$ independent truncated interferograms over the same exact fixed ground location. It corresponds to a unique state of the atmosphere that we seek to estimate from all the available measurements. Algorithmically speaking, using (Rodgers, 2000) notations and following his guidance in part 4.1.1, this can be achieved by including all $n$ NanoCarb truncated interferograms inside the same measurement vector $\mathbf{y} = [\mathbf{y_1}, \dots, \mathbf{y_n}]$ to retrieve one unique posterior state $\hat{\mathbf{x}}$. This posterior state maximizes the probability $P(x|\mathbf{y_1}, \dots, \mathbf{y_n})$ that can be expressed with Bayes theorem, and because measurements $\mathbf{y_i}$ are independent, as:

$$P(x|\mathbf{y_1}, \dots, \mathbf{y_n}) = \frac{P(\mathbf{y_1}, \dots, \mathbf{y_n}|x)P(x)}{P(\mathbf{y_1}, \dots, \mathbf{y_n})} = P(x) \prod_{i=1}^{n} \frac{P(\mathbf{y_i}|x)}{P(\mathbf{y_i})} \tag{A1}$$

Assuming Gaussian statistics for both state and measurements, and a linear forward model described by its jacobian matrices $\mathbf{K_i}$ corresponding to the measurement $\mathbf{y_i}$, we can express the a posteriori covariance matrix $\hat{\mathbf{S}}$ of the unique posterior state $\hat{\mathbf{x}}$ as:

$$\hat{\mathbf{S}} = \left[ \mathbf{S_a^{-1}} + \sum_{i=1}^{n} \mathbf{K_i^T S_{e,i}^{-1} K_i} \right]^{-1} \tag{A2}$$

with $\mathbf{S_a}$, the a priori covariance matrix of the a priori state vector $\mathbf{x_a}$, and $\mathbf{S_{e,i}}$, the a priori covariance matrix of the individual measurement $\mathbf{y_i}$. At the same time, for all individual a posteriori states $\hat{\mathbf{x}}_i$, retrieved from the individual independent measurements $\mathbf{y_i}$, their a posteriori covariance matrix $\hat{\mathbf{S}}_i$ can be expressed as:

$$\hat{\mathbf{S}}_i = \left[ \mathbf{S_a^{-1}} + \mathbf{K_i^T S_{e,i}^{-1} K_i} \right]^{-1} \tag{A3}$$

Thus, using Eq. (A2) and Eq. (A3), we can express $\hat{\mathbf{S}}$ as a function of individual a posteriori covariance matrices $\hat{\mathbf{S}}_i$:

$$\hat{\mathbf{S}}^{-1} = \mathbf{S_a^{-1}} + \sum_{i=1}^{n} (\hat{\mathbf{S}}_i^{-1} - \mathbf{S_a^{-1}}) \tag{A4}$$

Regarding the unique a posteriori state $\hat{\mathbf{x}}$, we have:

$$\hat{\mathbf{x}} = \mathbf{x_a} + \hat{\mathbf{S}} \sum_{i=1}^{n} \mathbf{K_i^T S_{e,i}^{-1}} (\mathbf{y_i} - \mathbf{K_i x_a}) \tag{A5}$$

and at the same time, individual a posteriori states $\hat{\mathbf{x}}_i$, retrieved from the individual independent measurements $\mathbf{y_i}$ also verify:

$$\hat{\mathbf{x}}_i = \mathbf{x_a} + \hat{\mathbf{S}}_i \mathbf{K_i^T S_{e,i}^{-1}} (\mathbf{y_i} - \mathbf{K_i x_a}) \tag{A6}$$

Thus, using Eq. (A5) and Eq. (A6), we can express $\hat{\mathbf{x}}$ as a function of individual a posteriori states $\hat{\mathbf{x}}_i$:

$$\hat{\mathbf{S}}^{-1}(\hat{\mathbf{x}} - \mathbf{x_a}) = \sum_{i=1}^{n} \hat{\mathbf{S}}_i^{-1}(\hat{\mathbf{x}}_i - \mathbf{x_a}) \tag{A7}$$

In conclusion, assuming all individual a posteriori state vectors $\hat{\mathbf{x}}_i$ are obtained with their respective posterior covariance matrices $\hat{\mathbf{S}}_i^{-1}$, Eq. (A4) and Eq. (A7) explain how to combine them in order to compute the unique posterior state $\hat{\mathbf{x}}$ and its covariance matrix $\hat{\mathbf{S}}$.

**Data availability**

Level 2 performance parameter files are available upon request from Matthieu Dogniaux by email (matthieu.dogniaux@lmd.ipsl.fr). The NanoCarb instrument model is available upon request from Silvère Gousset by email (silvere.gousset@univ-grenoble-alpes.fr). SPEX linear error analysis results are available upon request from Lianghai Wu by email (L.Wu@sron.nl). Finally, third-party datasets on which the L2 error parameterization can be applied are available upon request from Bojan Sic by email (Bojan.sic@noveltis.fr).

**Author contributions**

MD and CC carried out the L2 performance assessment and parameterization of the results. ELC, LC, SG and YF designed the NanoCarb concept; and SG developed the instrumental model with LC. LW and OH carried out the SPEXone linear error analysis. BS produced the ground tracks and auxiliary parameter datasets on which to apply L2 error parameterization. CC and LB supervised the work, and finally, MD wrote this article with feedbacks from all co-authors.

**Competing interests**

The authors declare that they have no conflict of interest.

**Financial support**

The Space CARBon Observatory (SCARBO) project received funding from the European Union's H2020 research and innovation program under grant agreement No 769032.

**Acknowledgements**

This work has received funding from CNES and CNRS. MD is funded by Airbus Defence and Space in the framework of a scientific collaboration with École polytechnique. NanoCarb initiated in the framework of the LabEx FOCUS ANR-11-LABX-0013. This work has been partly supported by a grant from Labex OSUG (Investissements d'avenir – ANR10 LABX56). The authors thank the whole SCARBO consortium for their help in preparing this paper.

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
