# Peer review of "The Space CARBon Observatory (SCARBO) concept: Assessment of XCO2 and XCH4 retrieval performance"

_Atmospheric Measurement Techniques, 2021_

## Author Comment (AC1)

**We would like to thank the referee for the kind feedback and relevant comments. All the points will be addressed in blue, between the referee's comments.**

The authors introduce a new concept called the Space CARBon Observatory (SCARBO), which aims to measure CO2 and CH4 from a constellation of ~20 satellites in sun-synchronous low Earth orbit, with a multi-angle polarimetric aerosol instrument to account for scattering effects. SCARBO will have higher spatial coverage and revisit frequency compared to existing greenhouse gas missions. The authors assess the performance of SCARBO for a variety of scenarios, both with and without the aerosol instrument. They find that systematic errors in column-averaged CO2 and CH4 (XCO2 and XCH4) retrievals can be greatly reduced by using aerosol information from the polarimeter. The authors also parameterize results as a function of relevant parameters in order to facilitate efficient computation of error maps for CO2 and CH4 flux estimation.

The manuscript is well written and the topic extremely relevant to the greenhouse gas remote sensing community. However, a few issues need to be addressed before it is ready for publication.

Lines 148-149: "Entanglements between CO2, CH4, O2, H2O and aerosols signals have been considered, with the assumption that albedo models are constant over all four spectral bands." What is the impact of varying albedo on the results? Also, only soil, vegetation and desert types are considered. What about water? Many emission sources (e.g., power plants) are near the ocean, so coastal areas would need to be considered.

**Regarding varying albedo models**

This sentence intent, at lines 148-149 (pre-print), is to explain the design hypotheses used for the NanoCarb concept version considered in this article. As it is the very first L2 performance assessment realised for the NanoCarb concept with a complete inverse radiative transfer scheme, the scope of this study lies within the current design hypotheses, and those will be challenged in upcoming studies as the NanoCarb concept gains maturity.

We adapted the revised manuscript at lines 153-156.

**What would happen if varying albedo models were considered?**

The narrow band filters, that select the incoming light to produce NanoCarb truncated interferograms, have central wavenumbers that shift towards larger wavenumbers with increasing incident angle with regard to the normal to the filter plane (pre-print line 469), i.e. with distance to the FOV center.

Thus, considering wavelength-dependant albedo models means that slightly different "effective" albedo values would have been used depending on the pixel position within the FOV (central symmetry according to its center) and on the strength of the albedo model wavelength-dependence (stronger for the VEG model than for SOL or DES). Consequently:
- it slightly changes the random error swath-dependence, due to slightly different baseline strength
- it results in perturbation of systematic errors if the wavelength-dependence cannot be perfectly retrieved (tests have shown that NanoCarb truncated interferograms currently – in the latest design – carry no-to-little information content on albedo wavelength-dependent slope) or if the wavelength-dependence is not perfectly known a priori (it is never the case at 2x2 $km^2$ resolution).

**Regarding water surface type**

The case of water has not been considered for this study because the NanoCarb instrument – as asked later, and indeed it was not mentioned in the pre-print – has been designed as a nadir-pointing instrument. Thus, it makes it impossible to track the specular reflection of the solar radiation over water, in order to get a satisfying signal-to-noise ratio over this surface type.

We have changed the sentence presenting NanoCarb in the introduction to directly state that it is a nadir-pointing instrument (revised manuscript line 81).

Lines 235-236: "The interfering impact of temperature has not been taken into account for the latest optimized OPD selection used in this work, and is not considered in the state vector." What is the impact of this assumption on the retrievals?

As for all CO2 observing concepts, knowledge of the atmospheric temperature profile is required for the correct retrieval of CO2 atmospheric concentration, and it is also the case for this NanoCarb concept. However, this interfering impact of temperature has not been taken into account for the optimized OPD selection (i.e. NanoCarb design) used in this study, thus temperature has not been included in the state vector.

Similarly to the answer to the previous question, the hypotheses used here for the L2 performance assessment are consistent with those used for designing the optimized OPD selection.

However, preliminary sensitivity tests (outside the scope of this article) have been conducted to prepare an upcoming iteration of the NanoCarb concept.

Including a global shift of the temperature profile in the state vector, with a standard deviation of 5K in the state a priori covariance matrix, results in an increase of XCO2 random error of about +0.25 ppm, consistent with the well-known entanglement between CO2 and temperature variables in inverse radiative transfer.

The impact on XCH4 random error is negligible, as CH4 is much less entangled with temperature variable. For instance TROPOMI - S5P ATBD, where temperature is optional in the state vector for its XCH4 product (page 26): https://sentinel.esa.int/documents/247904/2476257/Sentinel-5P-TROPOMI-ATBD-Methane-retrieval, and not included for the official XCH4 product (Lorente et al. (2021).

As this study focuses on scattering-related errors, the impact of atmospheric temperature misknowledge on systematic errors is not examined. We can however expect a perturbation of scattering-related systematic errors if we were to include a global shift of the temperature profile in the state vector, as the misknowledge of scattering-related variables would propagate differently through the averaging kernel matrix to CO2 and temperature related variables, which are somewhat entangled.

What is the impact of retrieving profile scaling factors for CO2 and CH4 as opposed to retrieving the vertical profile (that is traditionally done by OCO-2, for example)? Have the authors assessed the impacts on accuracy and on downstream flux estimation?

The choice of using a scaling factor to represent GHG variables in the state vector, rather than a profile like the ACOS algorithm, has been made after early preliminary tests that showed the low information content for GHG related variables (DOFs < 1), thus not requiring to offer – with a profile – the possibility to reach DOFs > 1, like the ACOS algorithm for OCO-2, for which CO2 related variables amount to DOFs ~2. Reducing, in this way, the representation of GHG in the state vector to a unique element per gas helped to better identify the entanglements between albedo and GHG variables.

As for the impacts of this choice, those can be exemplified by running a simple performance study for ALB=VEG, SZA=50°, CLH=2km, COD=0.02 and FOD=0.05. Let us consider a similar state vector as the one used in the article, but for CO2 related parameters. Let us consider two cases: (1) CO2 is represented as a profile scaling factor with an a priori random error equivalent to 13.95 ppm for XCO2 (2) CO2 is represented as a 19-layer vertical profile, with an a priori covariance matrix similar to ACOS, yielding an a priori random error of 13.95 ppm for XCO2. Table R1.1 gives the CO2 related DOFs and combined XCO2 random error for the transversal position at the center of the FOV.

**Table R1.1** DOFs and XCO2 random errors for two different representations of CO2 in the state vector

|                    | (1) Profile scaling factor | (2) 19-layer vertical profile |
|--------------------|----------------------------|-------------------------------|
| **DOFs**           | 0.86                       | 0.91                          |
| **XCO2 random error** | 0.56 ppm                | 0.45 ppm                      |

The implicit strong covariance between atmospheric levels in the scaling factor case offers slightly less degrees of freedom than when using a 19-layer vertical profile (ACOS covariance matrix used here), thus resulting in higher random error for the scaling factor case, compared to the vertical profile. Consequently, the results presented in the pre-print are conservative in information content and random errors.

This choice also has an impact on the averaging kernel shape, which is shown in Fig. R1.1. Using a profile instead of a scaling factor tends to diminish the vertical sensitivity values compared to when using a scaling factor. Above 1 values when retrieving a scaling factor are usual, as it can be seen for instance in Fig. 2. in Buchwitz et al. (2005).

[Figure]

**Figure R1.1** XCO2 averaging kernels in the case of retrieving a CO2 profile scaling factor (full line) and a 19-layer vertical profile (dashes).

The differences between profile scaling factor and layer profile retrievals undoubtedly lead to differences in systematic errors caused by a priori misknowledge of scattering particles. However, as the goal of this study was to realise the very first assessment of the SCARBO concept performance, and identify its forces and current limitations, we did not explore the impact of state vector design. Such questions will be addressed in later studies when the NanoCarb concept will have matured and its current limitations (see article conclusions) will be overcome.

The downstream flux estimation performance study only relies on the L2 parameterization which principle is detailed in Sect. 6 of the pre-print. Thus, the full impact of the state vector design on flux estimation performance has not been studied.

Lines 300-301: "For this synthetic performance study, constant trace gas concentration profiles have been used: 394.85 ppm for CO2 and 1855.3 ppb for CH4." This seems (unnecessarily) restrictive (see also previous comment). There needs to be an assessment of how results change for realistic CO2 and CH4 profiles.

This study presents the first performance assessment of the SCARBO concept, which is still being developed. This hypothesis of constant CO2 and/or CH4 profiles used for performance assessments is a usual one. For instance, the CarbonSat performance assessment study realised Buchwitz et al. (2013) – that inspired the method used for this SCARBO assessment – does not precise the a priori CO2 profile it uses, but appears to rely on the pre-existing work performed by Bovensmann et al. (2010), that assumed a constant vertical CO2 background.

However, we agree with the referee that realistic CO2 and CH4 profiles must eventually be considered for a final SCARBO evaluation study, that could be a full OSSE and that would demonstrate its maturity to fly and accomplish its mission.

We adapted the paragraph describing the a priori atmosphere to include these previous comments (revised manuscript line 313-315).

Aerosols: the authors might want to say that the fine mode particles are assumed to be spherical. It would also be useful to have a sentence describing how the aerosol single scattering properties were calculated (e.g., Mie for spherical, T-Matrix for spheroidal?).

Nonspherical aerosols are described as a size–shape mixture of randomly oriented spheroids (Hill et al., 1984; Mishchenko et al., 1997). We use the Mie- and T-matrix-improved geometrical optics database by Dubovik et al. (2006) along with their proposed spheroid aspect ratio distribution for computing optical properties for a mixture of spheroids and spheres.

We added more information (see revised manuscript lines 325-329).

Is SCARBO only going to make measurements in the nadir mode? If not, the viewing zenith angle needs to be a parameter that is considered in the evaluation of the scattering error.

As answered for the first comment of this review, only nadir measurements are considered for the SCARBO concept at this time. The introduction presenting the SCARBO concept was adapted accordingly in the revised manuscript.

**Grammatical Errors / Typos:**

**Thank you very much for catching these typos and English mistakes, we provide the line(s) in the revised manuscript where they have been corrected.**

Line 118: Acronym OPD already defined

We removed this redundant acronym definition (revised manuscript line 123)

Line 151: FOV (2) an analytical approximation -> FOV, and (2) an analytical approximation

We fixed the punctuation (revised manuscript line 158).

Lines 152, 258: "line-by-line" would be more appropriate than "pseudo-infinite"

We changed to "line-by-line" (revised manuscript lines 159, 268).

Lines 171-172: "The constellation sizing aims at ensuring intra-daily revisit of the largest possible amount of anthropogenic CO2 emission hotspots which emission rate is compatible with the 1 ppm SCARBO ð  ‹!!! precision objective." Awkwardly phrased

We revised this sentence into two sentences (revised manuscript line 178-179).

Line 173: "performed" -> "compiled"?

We changed the word (revised manuscript line 181).

Line 178: remove "a" before "global coverage" and "daily revisit"

We removed the articles (revised manuscript line 186).

Line 179: compromise well -> provides an optimal compromise

We changed the formulation (revised manuscript line 187-188).

Line 191: measures -> measurements

We changed the word (revised manuscript line 199).

Lines 197-198: what is meant by "without artificial noise"? The text indicates that instrument noise is considered in the retrievals. I would recommend removing this phrase to avoid confusion.

We mean that we do not add a random draw of artificial noise on these artificial NanoCarb truncated interferograms. The revised manuscript has been adapted to better reflect this idea (revised manuscript line 205-206): it is important to distinguish random perturbation of an artificial measurement with a noise model, and the accounting of the noise model in the Optimal Estimation framework.

Line 260: measure -> measurement

We changed the word (revised manuscript line 269).

Line 268: Acronym FOV already defined

We removed this redundant acronym definition (revised manuscript line 278).

Line 276: fasten -> speed up

We corrected this (revised manuscript line 287).

Line 368: "more disadvantageous" is too vague. Please use a more descriptive term.

We changed "more disadvantageous" to "lower" (revised manuscript line 385)

Lines 369-360: "more favourable" please use a more quantitative term (more forward scattering?)

Spaceborne aerosol instruments mostly sample the backscatter region of the aerosol phase function. Having large SZA helps to have a larger scattering angle range to sample the aerosol phase function, and it also helps to have this angle range closer to a 90° scattering angle, see Hasekamp et al. (2019). We adapted the text according to this explanation (revised manuscript line 386-387).

Line 448: of all atmospheric layers -> in all atmospheric layers

We corrected this mistake (revised manuscript line 466).

Line 498: on the optical path -> in the optical path

We corrected this mistake (revised manuscript line 516).

**References**

Lorente, A., Borsdorff, T., Butz, A., Hasekamp, O., aan de Brugh, J., Schneider, A., Wu, L., Hase, F., Kivi, R., Wunch, D., Pollard, D. F., Shiomi, K., Deutscher, N. M., Velazco, V. A., Roehl, C. M., Wennberg, P. O., Warneke, T., and Landgraf, J.: Methane retrieved from TROPOMI: improvement of the data product and validation of the first 2 years of measurements, Atmos. Meas. Tech., 14, 665–684, https://doi.org/10.5194/amt-14-665-2021, 2021.

Buchwitz, M., de Beek, R., Burrows, J. P., Bovensmann, H., Warneke, T., Notholt, J., Meirink, J. F., Goede, A. P. H., Bergamaschi, P., Körner, S., Heimann, M., and Schulz, A.: Atmospheric methane and carbon dioxide from SCIAMACHY satellite data: initial comparison with chemistry and transport models, Atmos. Chem. Phys., 5, 941–962, https://doi.org/10.5194/acp-5-941-2005, 2005.

Bovensmann, H., Buchwitz, M., Burrows, J. P., Reuter, M., Krings, T., Gerilowski, K., Schneising, O., Heymann, J., Tretner, A., and Erzinger, J.: A remote sensing technique for global monitoring of power plant CO2 emissions from space and related applications, Atmos. Meas. Tech., 3, 781–811, https://doi.org/10.5194/amt-3-781-2010, 2010.

Buchwitz, M., Reuter, M., Bovensmann, H., Pillai, D., Heymann, J., Schneising, O., Rozanov, V., Krings, T., Burrows, J. P., Boesch, H., Gerbig, C., Meijer, Y., and Löscher, A.: Carbon Monitoring Satellite (CarbonSat): assessment of atmospheric CO2 and CH4 retrieval errors by error parameterization, Atmos. Meas. Tech., 6, 3477–3500, https://doi.org/10.5194/amt-6-3477-2013, 2013.

Dubovik, O., et al. (2006), Application of spheroid models to account for aerosol particle nonsphericity in remote sensing of desert dust, J. Geophys. Res., 111, D11208, doi:10.1029/2005JD006619.

Hasekamp, O. P., Fu, G., Rusli, S. P., Wu, L., Di Noia, A., aan de Brugh, J., Landgraf, J., Martijn Smit, J., Rietjens, J. and van Amerongen, A.: Aerosol measurements by SPEXone on the NASA PACE mission: expected retrieval capabilities, J. Quant. Spectrosc. Radiat. Transf., 227, 170–184, doi:https://doi.org/10.1016/j.jqsrt.2019.02.006, 2019.

---

## Author Comment (AC2)

**The authors would like to thank the referee for the kind review and constructive comments. All the comments will be addressed in blue, between each question within the review text.**

The authors present a simulation-based assessment of CO2 and CH4 column retrievals for a novel nanosatellite constellation concept called SCARBO. The constellation would involve 20+ small satellites each carrying a Fabry-Perot interferometer (NanoCarb) for CH4 and CO2 and a multi-angle polarimeter (SPEXone) for aerosols. The authors present extensive error analysis with a focus on aerosol-related retrieval errors and the ability of the SPEXone auxiliary instrument to mitigate those errors. They use for this purpose standard OSSE methods including Rodgers (2000) optimal estimation techniques, and show that their NanoCarb-SPEXone instrument design should be capable of delivering high-precision XCO2 and XCH4 retrievals. The paper is well written and a good fit for AMT. I recommend acceptance for publication subject to the following comments and questions.

Specific comments

   L. 36: Please clarify what is meant by "on small areas".

We mean that urban areas, that concentrate 70% of fossil fuel related CO2 emission (Duren and Miller, 2012), are very small compared to the total continental surface, 0.5-0.6% of ice-free land surface, as reported in the latest IPCC WGIII report (https://report.ipcc.ch/ar6wg3/pdf/IPCC_AR6_WGIII_FinalDraft_Chapter08.pdf): 653,365 (Liu et al., 2020) km2 in 2015 compared to 130,000,000 km2 (IPCC, 2022) of ice-free land surface.

We adapted the introduction sentence to better reflect this idea (revised manuscript line 37).

   L. 40: Can you provide a reference for 2x2 km2 resolution being fine enough to resolve point sources? Of what magnitude? TROPOMI can resolve only extreme methane point sources at similar (5.5x7 km2) resolution (e.g., Pandey et al., 2019).

2x2 km2 is fine enough for all sources ≥10 MtCO2/year as per all the literature preparing CO2M mission: e.g. Bovensmann et al. (2010), Kuhlmann et al. (2019), etc. Higher spatial resolution is required to resolve smaller emission rates: e.g. 50x50 m2 for point sources ≥1 MtCO2/year (Strandgren et al., 2020).

We added the emission rate related to the 2x2 km2 spatial resolution, as well as Kuhlmann et al. (2019) reference in the introduction (revised manuscript line 42).

   L. 41: Extensive recent work has shown that plumes observed by imaging spectrometers do not look Gaussian. For example Cusworth et al. (2021) use an integrated mass enhancement method to quantify CO2 emissions from individual power plants observed by the PRISMA satellite instrument, and the TROPOMI team and others have used a variety of methods to quantify CH4 plume emission rates (eg, Pandey et al., 2019) at km-scale resolution, but Gaussian plume modeling seems poorly suited to the problem.

Indeed, there are many different methods relying on plume images. We adapted the formulation and added relevant references, including Varon et al. (2018) that notably compared three of these different approaches (revised manuscript lines 43-44).

   L. 66: "requirements for operational top-down monitoring of anthropogenic GHG emissions" – What are these requirements and what does "operational" CO2/CH4 monitoring mean?

Those requirements are: (1) imager instrument to resolve strong point source emission plumes (2) XCO2 precision better than 1 ppm, XCH4 precision better than 10 ppb (3) high revisit frequency, daily if possible. They were discussed for the design of the CarbonSat and then CO2M concepts, and identified as key to an European operational monitoring of fossil fuel CO2 emissions (Ciais et al., 2014, report by the EC: doi/10.2788/52148).

We adapted the text to reflect these details (see revised manuscript lines 69-71).

   L. 79: "geophysical parameters necessary to retrieve XCO2 and XCH4" – can you say what these parameters are or point to them in the text?

Those are some of the geophysical parameter involved in the Full Physics retrieval of XCO2 and XCH4. They include CO2, CH4, O2, H2O and aerosols, which have been taken into account for the OPDs optimisation that defined the current design of NanoCarb used for this study.

Surface albedo must also be taken into account and has been included in the state vector, as well as temperature which has not been included in the state vector here, as per current NanoCarb design hypotheses (see review #1 answers).

In order to lighten the already lengthy introduction, we just refer to the section where those aspects are presented (revised manuscript line 86).

L. 80-81: "2.3 x 2.3 km2 spatial resolution, enabling to detect emission plumes from megacities and hotspots (e.g. > 10 Mt CO2 yr-1 power plants)" – You seem to use "hotspot" and "point source" interchangeably. Megacities are examples of hotspots and power plants examples of point sources. For point sources, where does the 10 Mt/y threshold come from?

We corrected this sentence (revised manuscript line 87).

The 10 MtCO2/yr with a 2x2 km2 is the threshold that was determined for CO2M definition, based on resolution and XCO2 precision experiments. The SCARBO precision and resolution objectives were chosen to be close to those of CO2M.

References cited before in the text (e.g. Kuhlmann et al, 2019) already give details and reference papers regarding this threshold.

L. 86: Please describe CO2M.

The CO2M acronym is already defined at line 77 (revised manuscript) and the very complete Mission Requirement Document is cited right after: Meijer and Team, 2019.

L. 95: "scattering error-critical atmospheric and observational parameters" – Are these the "geophysical parameters" you mentioned before? It would be helpful to list these out somewhere in the introduction if not too lengthy or point to them in the text.

Those are not entirely the same: the 'geophysical parameters' mentioned before are parameters that influence radiative transfer in the shortwave infrared, whereas these 'scattering error-critical atmospheric and observational parameters' explore geophysical parameters (surface albedo model, aerosol layer optical depths and height) and an observational one (solar zenith angle).

Indeed, the following paragraph that announces the structure of the article, in the pre-print, did not mention these 'scattering-error-critical atmospheric and observational parameters'. It is fixed in the revised manuscript (revised manuscript lines 114-115).

Ray Nassar and Dan Cusworth's works about satellite monitoring of CO2 emissions from power plants should be cited somewhere. Same for TROPOMI methane plume papers (Pandey et al. and others) since there has been a lot of recent work on these topics.

Recent articles by Ray Nassar et al. (2021), Cusworth et al. (2021) and Pendey et al. (2019) have been added as references when mentioning plume emission rate methods earlier in the introduction (see revised manuscript lines 43-44).

Can you explain why SCARBO uses an FP rather than grating? Is it about financial cost, instrument size/weight, something else? Also please cite other instruments/concepts that use FP - eg Jervis et al. (2021).

The FP technology used by NanoCarb, for SCARBO, has been chosen to propose a very compact instrument, of a volume comprised in a few $cm^3$ compared to several tens of litres for conventional instruments. Gratings permit to reach high levels of performance, but the bulk silicon Fabry-Perot enables to build a smaller – and simpler – instrument.

Reference to GHGSat FP technology (Jervis et al., 2021) has been added when mentioning small satellite concepts (revised manuscript line 74-75).

L. 165: Not clear what "0.003" means, is it an error (1 or 2 sigma)?

0.003 comprises both systematic and random errors, and is used for the SPEXone performance assessment, thus equivalent to 1 sigma, see Hasekamp et al. (2019).

We added a parenthesis to better underline this (revised manuscript line 174).

L. 173: What is an "emission clump"? This terminology is non-standard.

Emission clumps are defined in Wang et al. (2019) paper: 'In this study, we characterize area and point fossil fuel CO2 emitting sources which generate coherent XCO2 plumes that may be observed from space. We characterize these emitting sources around the globe and they are referred to as "emission clumps" hereafter.'

We adapted the manuscript text to reflect this definition (see revised manuscript lines 181-182).

L. 234: Are you referring to the instrument temperature?

We are referring to atmospheric temperature; the revised manuscript has been adapted to provide this explanation (see revised manuscript lines 244).

Table 2: How conservative is the 4 hPa (0.4%) error for surface pressure? It seems quite small.

This is the uncertainty used in the ACOS algorithm that produces the official OCO-2 XCO2 product. See ATBD p 43: https://docserver.gesdisc.eosdis.nasa.gov/public/project/OCO/OCO_L2_ATBD.pdf

The same surface pressure a priori uncertainty was used for the 5AI inverse scheme paper Dogniaux et al. (2021).

L. 271: I would suggest pointing to the appendix here because I initially wondered if the "combination" of single-pixel measurements was through averaging or something more.

We moved the reference to the Appendix there (see revised manuscript lines 282).

L. 289-290: "Errors arising from the interpolation have been assessed and are negligible (not shown)" – What is the magnitude of the error?

For seven $\theta_T$ transversal positions within the swath ($\theta_T$ = -9.3°, -7.0°, -4.7°, 0.1°, +4.7°, +7.0°, +9.3°), for a situation corresponding to ALB=VEG, SZA=50°, CLH=2km, COD=0.15 and FOD=0.08, we evaluated the interpolation approximation errors between combined exactly retrieved L2 results and combined interpolated L2 results. If interpolation approximation errors can be seen between pixel-wise exactly retrieved results (black crosses in Fig. R2.1 and Fig. R2.2) and pixel-wise interpolated results (coloured lines in Fig. R2.1 and Fig. R2.2), the interpolation approximation errors become small to even negligible when pixel-wise results are combined (see Fig. R2.1 and Fig. R2.2): up to a maximum of 0.01 ppm for XCO2 systematic and random errors, up to a maximum of 0.05 ppb for XCH4 systematic and random errors.

We purposefully did not include this discussion in order to lighten the content of the paper, so we adjusted the revised manuscript just by adding a parenthesis explaining the magnitude of these errors (revised manuscript lines 301-302).

[Figure]

[Figure]

**Figure R2.1** Comparison of pixel-wise interpolated (coloured full lines) and exactly retrieved (black crosses) XCO2 systematic (top) and random (bottom) errors for 7 different transversal positions θ_T.

[Figure]

**Figure R2.2** Comparison of pixel-wise interpolated (coloured full lines) and exactly retrieved (black crosses) XCH4 systematic (top) and random (bottom) errors for 7 different transversal positions θ_T.

L. 297-300: CH4 falls off rapidly above the troposphere, so why use a uniform vertical profile? Is the impact of this unrealistic profile on the retrieval small enough to be neglected?

As it is the very first performance assessment of the – still maturing – NanoCarb concept realised by using a complete inverse radiative transfer scheme, we chose simplified CO2 and CH4 profiles. However, we agree with both referees that realistic CO2 and CH4 profiles must eventually be considered for a final SCARBO evaluation study, that could be a full OSSE and that would demonstrate its maturity to fly and accomplish its mission.

We adapted the paragraph describing the a priori atmosphere to include these previous comments (revised manuscript line 313-315).

Besides, several aspects of the study conducted here argue in favour of a small impact of this hypothesis:
-   we only consider scattering-related a priori misknowledge, the CH4 profile is supposed to be a priori perfectly known, thus no smoothing error can arise from a unrealistic CH4 profile in the retrievals
-   XCH4 column averaging kernels show a small sensitivity to the top of the atmosphere (AKs values of about 0.5, cf Fig. 8), arguing for little CH4 information coming from the top of the atmosphere.

Fig. 7: Is the much lower DOFs for FOD in the with-SPEX scenario merely due to using a much lower prior error for FOD compared to no-SPEX?

The much lower FOD DOFs in the with-SPEX case is – according to our understanding – the result of (1) the tight constraint brought by much lower a priori error brought by SPEXone (2) the already existing low information content regarding this mode of aerosols.

In other words, the information brought by SPEXone is so large compared to the one available in NanoCarb measurements (see larger but still low no-SPEX FOD DOFs) that only the a priori information contributes to the estimation of fine mode aerosols in the with-SPEX case. This is illustrated by the very low FOD DOFs in the with-SPEX case.

L. 424-425: Do the albedo DOFs actually increase? They seem to be equal to 1 in both scenarios.

Yes, they do. Albedo DOFs are slightly below 1, but it is not distinguishable at this scale. However, we recognize that this increase between no-SPEX and with-SPEX is very small and not distinguishable.

We adapted the manuscript to stress that, in the case of albedo, the increase is very small (revised manuscript line 443-444).

L. 437-439: This seems odd since the albedo DOFs look to be almost or exactly 1.0.

Albedo DOFs are indeed not exactly 1.0 (undistinguishable). We adapted the revised manuscript (revised manuscript line 419).

Another way to look at this estimation entanglement is to examine covariance coefficients between CO2 scaling factor and albedo parameters in the Optimal Estimation a posteriori covariance matrix. Divided by the standard deviations of these variables, it yields correlation coefficients. For example, for NanoCarb, we find a correlation coefficient of 0.72 between CO2 scaling factor and B2 albedo. By comparison, this correlation coefficient is 0.04 if run a similar exercise with an OCO-2 measurement.

Fig. 8: It's not clear how you compute column averaging kernels when your state vector doesn't include a vertical column but rather a single scaling factor for each gas. If the state vector included CH4 and CO2 at different vertical levels then you would obtain A = dxhat/dx giving the AK for each vertical layer. How do you get column averaging kernels when optimizing just a scaling factor?

We make use of a complementary way to compute the AK matrix: A = GK. The 5AI inverse scheme stores the atmospheric layer-wise GHG jacobians and uses them to compute a vertical column averaging kernel. Those are obtained with the following equation:
$$(a_{GHG})_j = G_{GHG} \, K_{GHG,j} \, x_{a,GHG,j} / h_j$$
with $(a_{GHG})_j$, the column averaging kernel of the j-th atmospheric layer for a given GHG, $G_{GHG} \, K_{GHG,j}$, the [1,m]x[m,1] matrix product between the GHG scaling factor related line of gain matrix $G_{GHG}$, and the stored j-th atmospheric layer GHG jacobian $K_{GHG,j}$ (with m being the length of the measure vector), $x_{a,GHG,j}$ the a priori

GHG concentration for the j-th atmospheric layer and $h_j$ the pressure weighting function of the j-th atmospheric layer.

Fig. 8: Also the column averaging kernels look quite smooth, what is the vertical resolution here?

The vertical resolution can be obtained by analysing the averaging kernel row peak widths, as per Rodgers (2000). While the equation written above enables to compute the column averaging kernel – through the gain matrix row dedicated to GHG scaling factor – the layer-wise averaging kernels are not available.

A simple way to try to analyse those is to realise a small performance test that considers a GHG profiles instead of scaling factors in the state vector. Fig R2.3 shows the layer-wise averaging kernels that we obtain for CO2, considering the a priori covariance matrix used by ACOS, the official OCO-2 algorithm. As seen on Fig R2.3, no width can be properly identified, which is consistent with the below-unity GHG degrees of freedom for NanoCarb measurements.

[Figure]

**Figure R2.3** Layer-wise CO2 averaging kernel obtained when retrieving a CO2 profile from NanoCarb measurements.

Several plots show regions where retrievals did not converge satisfactorily but I cannot find in the manuscript what method you use for the optimization. Is it Newton, Levenberg-Marquardt, something else?

We used Levenberg-Marquardt optimization method. We modified the revised manuscript to add this information (revised manuscript line 217-218).

Fig. 13: Why is there striping in this figure? Because of the loss of precision with increasing transversal position?

Indeed, the stripping is caused by the increase of random error with the transversal position within the swath. We modified the revised manuscript to add this explanation (revised manuscript lines 674-675).

Technical corrections

L. 39: "large-swath"

We corrected it (revised manuscript line 41).

L. 51: "best fits"

We corrected it (revised manuscript line 54).

L. 151: "radiance spectrum"

We corrected it (revised manuscript line 159).
L. 164: "at 50 spectral band" seems like a typo?

This is not a typo.

L. 178: "compromises"

We corrected it (revised manuscript lines 187-188).

L. 276: "fasten" doesn't seem like the right word here. Do you mean "speed up" or something similar?

We corrected it (revised manuscript lines 287).

L. 422: "in these situations" typo

We corrected it (revised manuscript lines 440).

L. 612: "then" typo

We corrected it (revised manuscript lines 631).

L. 637: "mentioned" typo

We corrected it (revised manuscript lines 656).

Consider changing "scattering error-critical" to "scattering-error-critical" everywhere. If I understand correctly it's meant to be a compound adjective and might be clearer with two –'s.

We made this change.

References

Pandey, S., Gautam, R., Houweling, S., van der Gon, H. D., Sadavarte, P., Borsdorff, T., Hasekamp, O., Landgraf, J., Tol, P., van Kempen, T., Hoogeveen, R., van Hees, R., Hamburg, S. P., Maasakkers, J. D., and Aben, I.: Satellite observations reveal extreme methane leakage from a natural gas well blowout, P. Natl. Acad. Sci. USA, 116, 26376–26381, https://doi.org/10.1073/pnas.1908712116, 2019.

D.H. Cusworth, R.M. Duren, A.K. Thorpe, M.L. Eastwood, R.O. Green, P.E. Dennison, C. Frankenberg, J.W. Heckler, G.P. Asner, C.E. Miller: Quantifying global power plant carbon dioxide emissions with imaging spectroscopy, AGU Adv., 2 (2) (2021), https://agupubs.onlinelibrary.wiley.com/doi/abs/10.1029/2020AV000350

Jervis, D., McKeever, J., Durak, B. O. A., Sloan, J. J., Gains, D., Varon, D. J., Ramier, A., Strupler, M., and Tarrant, E.: The GHGSat-D imaging spectrometer, Atmos. Meas. Tech., 14, 2127–2140, https://doi.org/10.5194/amt-14-2127-2021, 2021.

**References**

Liu, X., Huang, Y., Xu, X. et al. High-spatiotemporal-resolution mapping of global urban change from 1985 to 2015. Nat Sustain 3, 564–570 (2020). https://doi.org/10.1038/s41893-020-0521-x

Bovensmann, H., Buchwitz, M., Burrows, J. P., Reuter, M., Krings, T., Gerilowski, K., Schneising, O., Heymann, J., Tretner, A. and Erzinger, J.: A remote sensing technique for global monitoring of power plant CO2 emissions from space and related applications, Atmos. Meas. Tech., 3(4), 781–811, doi:10.5194/amt-3-781-2010, 2010.

Kuhlmann, G., Broquet, G., Marshall, J., Clément, V., Löscher, A., Meijer, Y. and Brunner, D.: Detectability of CO2 emission plumes of cities and power plants with the Copernicus Anthropogenic CO2 Monitoring (CO2M) mission, Atmos. Meas. Tech., 12(12), 6695–6719, doi:10.5194/amt-12-6695-2019, 2019.

Strandgren, J., Krutz, D., Wilzewski, J., Paproth, C., Sebastian, I., Gurney, K. R., Liang, J., Roiger, A. and Butz, A.: Towards spaceborne monitoring of localized CO2 emissions: an instrument concept and first performance assessment, Atmos. Meas. Tech., 13(6), 2887–2904, doi:10.5194/amt-13-2887-2020, 2020.

Ciais, P., Dolman, A. J., Bombelli, A., Duren, R., Peregon, A., Rayner, P. J., Miller, C., Gobron, N., Kinderman, G., Marland, G., Gruber, N., Chevallier, F., Andres, R. J., Balsamo, G., Bopp, L., Bréon, F.-M., Broquet, G., Dargaville, R., Battin, T. J., Borges, A., Bovensmann, H., Buchwitz, M., Butler, J., Canadell, J. G., Cook, R. B., DeFries, R., Engelen, R., Gurney, K. R., Heinze, C., Heimann, M., Held, A., Henry, M., Law, B., Luyssaert, S., Miller, J., Moriyama, T., Moulin, C., Myneni, R. B., Nussli, C., Obersteiner, M., Ojima, D., Pan, Y., Paris, J.-D., Piao, S. L., Poulter, B., Plummer, S., Quegan, S., Raymond, P., Reichstein, M., Rivier, L., Sabine, C., Schimel, D., Tarasova, O., Valentini, R., Wang, R., van der Werf, G., Wickland, D., Williams, M., and Zehner, C.: Current systematic carbon-cycle observations and the need for implementing a policy-relevant carbon observing system, Biogeosciences, 11, 3547–3602, https://doi.org/10.5194/bg-11-3547-2014, 2014.

Wang, Y., Ciais, P., Broquet, G., Bréon, F.-M., Oda, T., Lespinas, F., Meijer, Y., Loescher, A., Janssens-Maenhout, G., Zheng, B., Xu, H., Tao, S., Gurney, K. R., Roest, G., Santaren, D. and Su, Y.: A global map of emission clumps for future monitoring of fossil fuel CO2 emissions from space, Earth Syst. Sci. Data, 11(2), 687–703, doi:10.5194/essd-11-687-2019, 2019.

Dogniaux, M., Crevoisier, C., Armante, R., Capelle, V., Delahaye, T., Cassé, V., De Mazière, M., Deutscher, N. M., Feist, D. G., Garcia, O. E., Griffith, D. W. T., Hase, F., Iraci, L. T., Kivi, R., Morino, I., Notholt, J., Pollard, D. F., Roehl, C. M., Shiomi, K., Strong, K., Té, Y., Velazco, V. A. and Warneke, T.: The Adaptable 4A Inversion (5AI): description and CO2 retrievals from Orbiting Carbon Observatory-2 (OCO-2) observations, Atmos. Meas. Tech., 14(6), 4689–4706, doi:10.5194/amt-14-4689-2021, 2021.

Hasekamp, O. P., Fu, G., Rusli, S. P., Wu, L., Di Noia, A., aan de Brugh, J., Landgraf, J., Martijn Smit, J., Rietjens, J. and van Amerongen, A.: Aerosol measurements by SPEXone on the NASA PACE mission: expected retrieval capabilities, J. Quant. Spectrosc. Radiat. Transf., 227, 170–184, doi:https://doi.org/10.1016/j.jqsrt.2019.02.006, 2019.